# Antimicrobial protein REG3A regulates glucose homeostasis and insulin resistance in obese diabetic mice

Patrick Gonzalez[1,2], Alexandre Dos Santos[1,2], Marion Darnaud [1,2], Nicolas Moniaux [1,2], Delphine Rapoud[1,2], Claire Lacoste [1,2], Tung-Son Nguyen[1,2], Valentine S. Moullé[3], Alice Deshayes [1,2], Gilles Amouyal[4], Paul Amouyal[4], Christian Bréchot[5], Céline Cruciani-Guglielmacci[3], Fabrizio Andréelli[6], Christophe Magnan [3] & Jamila Faivre [1,2,7✉]

Innate immune mediators of pathogen clearance, including the secreted C-type lectins REG3 of the antimicrobial peptide (AMP) family, are known to be involved in the regulation of tissue repair and homeostasis. Their role in metabolic homeostasis remains unknown. Here we show that an increase in human REG3A improves glucose and lipid homeostasis in nutritional and genetic mouse models of obesity and type 2 diabetes. Mice overexpressing REG3A in the liver show improved glucose homeostasis, which is reflected in better insulin sensitivity in normal weight and obese states. Delivery of recombinant REG3A protein to leptin-deficient ob/ob mice or wild-type mice on a high-fat diet also improves glucose homeostasis. This is accompanied by reduced oxidative protein damage, increased AMPK phosphorylation and insulin-stimulated glucose uptake in skeletal muscle tissue. Oxidative damage in differentiated C2C12 myotubes is greatly attenuated by REG3A, as is the increase in gp130-mediated AMPK activation. In contrast, Akt-mediated insulin action, which is impaired by oxidative stress, is not restored by REG3A. These data highlight the importance of REG3A in controlling oxidative protein damage involved in energy and metabolic pathways during obesity and diabetes, and provide additional insight into the dual function of host-immune defense and metabolic regulation for AMP.

[1] INSERM, U1193, Paul-Brousse University Hospital, Hepatobiliary Centre, Villejuif 94800, France. [2] Université Paris-Saclay, Faculté de Médecine Le Kremlin-Bicêtre, Le Kremlin-Bicêtre 94270, France. [3] Université of Paris, Unité de Biologie Fonctionnelle et Adaptative, CNRS UMR 8251, Paris 75013, France. [4] Alfact Innovation, Paris 75001, France. [5] USF Health, University of South Florida, Tampa, USA. [6] Sorbonne Université, INSERM, NutriOmics team, Institute of Cardiometabolism and Nutrition (ICAN), Assistance Publique-Hôpitaux de Paris, Pitié-Salpêtrière Hospital, Paris 75013, France. [7] Assistance Publique-Hôpitaux de Paris (AP-HP). Université Paris Saclay, Medical-University Department (DMU) Biology, Genetics, Pharmacy, Paul-Brousse Hospital, Villejuif 94800, France. ✉email: jamila.faivre@inserm.fr

Metabolic syndrome is a serious health condition that affects 25-35% of American and European adults[1,2], and is characterized by abdominal obesity, insulin resistance, hypertension, and hyperlipidemia resulting from the increasing prevalence of obesity. This contributes to the pathogenesis of multiple chronic diseases, such as type 2 diabetes mellitus (T2D), non-alcoholic fatty liver disease (NAFLD), cardiovascular diseases, stroke, and cancers. Patients with metabolic disorders may also face a higher risk for various failures in the body's defenses, including an increased risk of infection, changes in immune monitoring and modulation, lack of activation of T cells that have become resistant to insulin in a context of low-grade inflammation and oxidative stress.

Immune-endocrine interrelationships represent a major circuitry for maintaining proper homeostasis and energy metabolism, this cross-organ communication being highly conserved throughout evolution. A large body of evidence indicates functional links between metabolic control with the innate immune system, the primary and most ancient defense against infection[3]. For example, the activation of NF-κB by a pathogen sensing system (Toll-like receptors) or the tumor necrosis factor (TNF) signaling blocks systemic insulin action and glucose metabolism in response to nutrient deprivation or sepsis as an adaptive response to reallocate energy use to the affected organs whose acute requirements need to be met[4,5]. Antimicrobial peptides and proteins (AMPs), which are fundamental effectors of the innate immune response by exerting microbicidal activities against multiple types of microorganisms, are components of the immune-metabolic crosstalk. They include defensins, C-type lectins of the REG3 family, and cathelicidins. The expression of a number of them is finely modulated by nutrients and metabolic hormones, such as the adipokines visfatin and leptin[6-8]. Some AMPs such as cathelicidin LL37/CRAMP and alpha-melanocyte-stimulating hormone α-MSH have been implicated in energy and glucose regulation by stimulating insulin production or inhibiting hepatic gluconeogenesis, respectively[9-12]. This suggests a paradigm shift towards a dual function of innate host defense and metabolism for AMP, which, unlike pattern recognition receptors, are secreted molecules that can act at a distance from their production site to ensure metabolic adaptation of the organism to the environment and nutrition.

In the intestine, human regenerating islet-derived protein 3-alpha (REG3A), also known as hepatocarcinoma-intestine-pancreas/pancreatic associated protein or HIP/PAP, is a secreted 16 kDa carbohydrate-binding protein that functions as an AMP against Gram-positive bacteria[13]. Antibacterial activity requires N-terminal proteolytic processing by trypsin or elastase of intestinal REG3 proteins generating a 15 kDa protein. This cleavage does not occur for the extra-intestinal and circulating forms of REG3A[14-16]. REG3A also helps shape the symbiotic interactions of the human host and the gut microbiota through an antioxidant capacity to scavenge toxic reactive oxygen species (ROS), thus preserving the bacterial symbionts and the mucosal barrier[17]. In tissues, the action of REG3 goes beyond its activity towards microbes. REG3 genes are expressed in a limited number of tissues and cell types under physiological conditions, and in greater numbers during tissue damage and inflammation particularly in response to interleukin-6 (IL6) family cytokines, where they promote tissue repair and healing. These undoubtedly positive actions of REG3 proteins have been reported in vivo in several internal tissues and in the skin[17-26]. The mechanisms by which REG3A initiates intracellular signaling remain poorly defined, and involve, depending on the context, tissue and cell type, carbohydrate ligand recognition, carbohydrate-protein interaction, or protein-protein interaction. Several potential surface receptors and membrane-associated proteins (such as glycoprotein 130 (gp130), fibronectin, epidermal growth factor

receptor (EGFR), exostosin-like glycosyltransferase 3 (EXTL3), syndecan2, annexin3) have been proposed to interact with REG3A and transmit the REG3A signal into cells[27-33]. This report deals with the action of REG3A on type 2 diabetes mellitus and obesity which remains largely unknown, being mainly studied in type 1 diabetes in humans and experimental animals, as a potential inducer of protection and regeneration of pancreatic beta cells[34-46]. Our findings reveal a previously unknown mechanism by which REG3A regulate glucose homeostasis by combating oxidative stress and activating AMPK in skeletal muscle, and suggest an additional level of regulation of metabolism by lectin-mediated innate immunity.

## Results

**Hepatic expression of human REG3A decreases adiposity in aged mice.** We have begun to explore the metabolic effects of REG3A by characterizing the metabolic phenotype of healthy and diabetic C57BL/6 mice overexpressing human REG3A. We used previously generated homozygous transgenic C57BL/6 mice expressing human REG3A in hepatocytes (TG-REG3A) under the control of the mouse albumin gene promoter[47] and fed them a regular diet (51.7% carbohydrates, 21.4% protein, 5.1% fat). Transgenic hepatocytes secreted REG3A into blood vessels through the basolateral membranes, resulting in basal serum REG3A levels of $242 \pm 14$ ng/mL in females ($n = 50$) and $316 \pm 17$ ng/mL in males ($n = 50$). This allowed us to study the action of highly concentrated circulating REG3A on metabolic homeostasis beyond the impact on liver as this extrahepatic action is currently largely unknown. Although weight differences existed between TG-REG3A mice and wild-type (WT) mice, they were modest and were observed in females rather than males. In female TG-REG3A mice, weight gain was lower than in WT mice at 4, 6, and 18 months of age ($21.6 \pm 3.4$ g vs. $22.8$ g $\pm 1.6$ g, $P = 5.10{-4}$; $24.7 \pm 2.6$ g vs. $25.2 \pm 6.6$ g, $P = 0.032$; $28.2 \pm 3.8$ g vs. $30.1 \pm 4.4$ g; $P = 7.10{-3}$, respectively). No significant difference was observed at 12 and 24 months (Supplementary Figure 1). TG-REG3A male mice showed no significant difference in weight compared with WT male mice until 18 months of age, after which a difference was observed at 24 months of age ($32.2$ g $\pm 5.4$ g vs. $33.8 \pm 3.9$ g, respectively; $P = 0.04$) (Fig. 1a). We selected male REG3A-transgenic mice ($n = 8$) and control mice ($n = 8$) at 4 months of age and studied their phenotypes over time. At this time, body weight was $29.0 \pm 0.6$ g and $28.0 \pm 0.6$, respectively ($P = 0.34$). We monitored body composition by Magnetic Resonance Imaging (MRI) and found that TG-REG3A mice showed a reduction in fat mass compared with WT mice at all times in the 2-year analysis ($-21\%$, $-59\%$, $-40\%$, $-42\%$, and $-43\%$ at 4, 6, 12, 18, and 24 months, respectively) with a concomitant increase in lean mass at 6, 12, 18 months (Fig. 1b, c). These differences in body composition were not explained by reduced food intake (Fig. 1d and Supplementary Fig. 2a). In fact, cumulative food intake was even higher ($+20\%$) in TG-REG3A than WT mice at 6 and 24 months (Fig. 1d and Supplementary Fig. 2a). In addition, bomb calorimetry measurements showed no defect in intestinal energy and macronutrient absorption in TG-REG3A mice compared with WT mice (Fig. 1e-h). Lastly, the spontaneous locomotor activity and energy expenditure were also similar between the two groups (Fig. 1i, j). Indirect calorimetry measurements of oxygen consumed (VO2) and carbon dioxide produced (VCO$_2$) yielded a higher respiratory exchange ratio (RER; VCO$_2$/VO$_2$) in TG-REG3A mice than in WT mice at several time points, indicating that carbohydrates rather than fat are oxidized during the dark (active) phase for energy production. In particular, we observed an accelerated transition from beta-oxidation to glycolysis. During the normal light phase, the RER of TG-REG3A mice

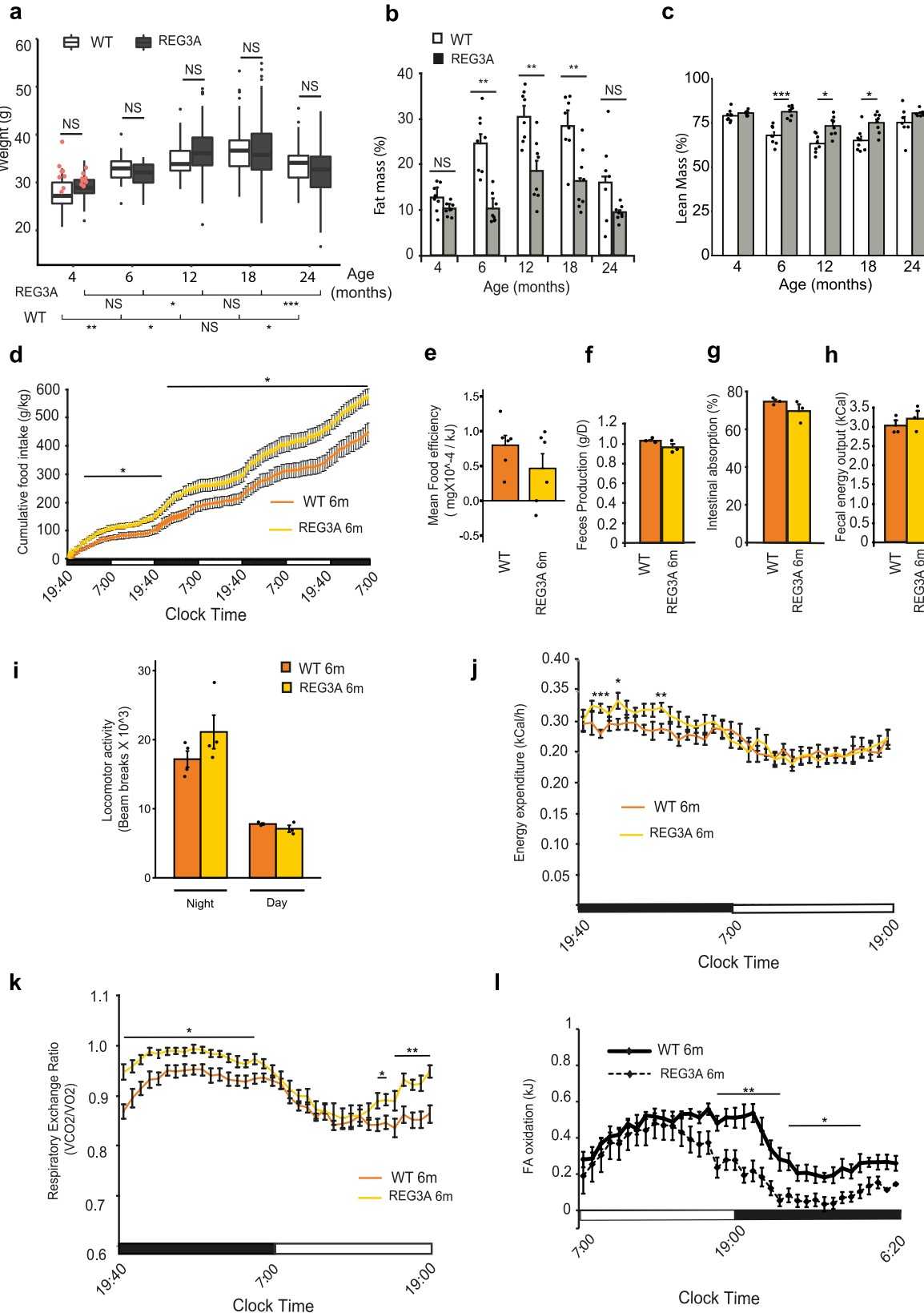

increased more rapidly than that of WT mice, suggesting that TG-REG3A mice preferentially use carbohydrate rather than fat during this period (Fig. 1k, l and Supplementary Fig. 2b, c). This difference persisted at 24 months and indicates the ability of transgenic mice to use glucose rather than fat during the fasting period throughout their lives.

**Increased sensitivity to insulin in REG3A transgenic mice fed a regular diet**. To determine whether the above reduction in body fat and increase in glucose oxidation in TG-REG3A mice are accompanied by specific metabolic traits, we examined glucose levels under basal conditions and in insulin and glucose tolerance tests. Levels of fasting blood glucose was lower in 6-month old

**Fig. 1 Suppression of body fat accumulation in transgenic mice expressing human REG3A in the liver. a** Changes in body weight in male REG3A-transgenic mice (REG3A, $n = 73$) and littermate wild-type (WT) controls ($n = 53$) fed with *ad libitum* standard chow (2830 kcal/kg). The difference between WT and REG3A groups is not significant ($P = $ NS) except 24 months (two-sided Wilcoxon rank sum test). Red dots: 16 randomly selected 4-month old mice (8 WT, 8 REG3A) for further metabolic characterization. **b–l** Metabolic phenotype of the 16 selected mice mentioned above. **b** Fat and (**c**) lean mass in WT ($n = 8$) and REG3A mice ($n = 8$) over time. **d–j** Feeding behavior, activity and metabolic performance of the same 8 WT and 8 REG3A mice at the age of 6 months. **d** Cumulative food intake over a 3.5-day period. **e** Food efficiency as calculated by the ratio of body weight gain over food ingested in kJ. **f** Feces produced per day. **g** Intestinal absorption as the ratio of amount of food intake over fecal output within a 24 h period. **h** Energy content in feces measured with a bomb calorimeter. **i** Locomotor activity averaged over two 12 h period. **j** Energy expenditure measured in a calorimetric chamber. **k** Respiratory quotient ($VCO_2/VO_2$) during a day-night cycle. **l** Fatty acid (FA) oxidation measured by respirometry over a day-night cycle. Data are averages ± SEM. *$p < 0.05$; **$p < 0.01$; ***$p < 0.005$ by Anova test (**a–c**) or two-way repeated measures Anova (**d**, **j–l**) both followed by Tukey post hoc multiple comparison test. NS or no statistical indication, no significance.

TG-REG3A chow-fed mice compared to age-matched WT (WT, $100 \pm 6$ mg/dL; REG3A, $77 \pm 3$ mg/dL) and equivalent in the two groups of mice at 24 months (WT, $115 \pm 5$ mg/dL; REG3A, $120 \pm 7$ mg/dL, Fig. 2a). In response to a glucose challenge (2 g/kg), 6-month-old TG-REG3 mice exhibit mild glucose intolerance, but 24-month-old mice do not (Fig. 2b); their peak glucose values are higher and their clearance times longer compared with control mice ($248 \pm 10$ vs. $190 \pm 5$ mg/dL at 30 min, and $230 \pm 12$ vs. $177 \pm 8$ mg/dL at 45 min, $p < 0.01$). At 60 min, glucose levels were no longer different from those of control mice. This was associated with a decrease in insulin output at the 30-min time point after glucose gavage. It should be noted that 24-month-old TG-REG3A mice with normal glucose tolerance showed a reduction in insulin output at times 0, 15, and 30 min after gavage, in contrast to the increase observed in aged WT mice (Fig. 2c). Plasma C-peptide was lower at baseline in TG-REG3A mice than in WT mice at 6 and 24 months. In both mice and at both ages, it was elevated in response to glucose, indicating similar responsiveness of their endocrine pancreas (Supplementary Fig. 3a). According to the homeostasis model assessment of insulin resistance (HOMA-IR), TG-REG3 mice showed lower HOMA scores than WT mice, indicating that TG-REG3 mice improved, and maintained their better insulin sensitivity with advanced age (Fig. 2d). Of note, 24-month-old WT mice show a trend toward basal hyperinsulinemia compared with 6-month-old WT mice ($p = 0.17$), which is reflected by the increase in the HOMA-IR score (Fig. 2c, d). This pattern of basal hyperinsulinemia in aged WT mice combined with relatively preserved insulin sensitivity (Fig. 2e) is consistent with previously reports in elderly human subjects[48] and 18-month-old C57bl/6 mice[49]. In an insulin tolerance test (0.75 units/kg), TG-REG3 mice became more hypoglycemic after insulin injection than WT mice at 6 and 24 months (Fig. 2e). We analyzed the contribution of liver, adipose tissue, and skeletal muscle in improving insulin sensitivity in TG-REG3A mice by assessing the level of Akt phosphorylation at Ser473 in these tissues. Compared with WT mice, increased phosphorylation of Akt on Ser473 was observed in the liver of TG-REG3A mice at 6 months (but not at 24 months) of age, whereas no differences were found in subcutaneous white adipose tissue (WAT), tibialis anterior, and soleus muscles (Supplementary Fig. 3b, c).

**Improved insulin sensitivity and reduced hyperinsulinemia in REG3A transgenic mice fed a high fat diet.** To determine whether obese TG-REG3A mice had improved whole-body glucose homeostasis, they were fed a high-fat diet (HF260; 5505 kcal/kg) for 8 weeks, in which 60% of energy came from fat and 27% from carbohydrate. We found that females were more resistant to HFD-induced obesity and metabolic alteration than males, in agreement with previous report[50]. Glucose and insulin tolerance were better in females than in males ($P = 0.004$ for OGTT and $P = 3.10-4$ for ITT; two-way Anova). This better insulin sensitivity of females is comparable regardless the genotype is WT or

transgenic (Supplementary Fig. 4). We also found that systemic insulin sensitivity did not correlate with circulating REG3A levels in either the male or female groups, suggesting that changes in circulating levels do not affect the extent of insulin sensitivity control. We then proceeded with male mice with clearly established metabolic disease. The amount of food ingested was similar in WT and REG3A transgenic mice throughout the diet (Fig. 3a). Weight gain was also similar in WT and REG3 transgenic mice with weight increasing by 55-60% during this period ($44.9 \text{ g} \pm 2.7$ g in WT vs. $42.7 \text{ g} \pm 9.7$ g in REG3A-TG) (Fig. 3b). Basal fasting blood glucose after two months of high-fat feeding did not differ between the two groups of mice ($220 \pm 7$ mg/dL in WT; $224 \pm 8$ mg/dL in REG3A-TG). In response to oral glucose, the blood glucose escalation curves were largely similar between the two groups of mice, indicating similar glucose tolerance under HF260 (Fig. 3c). The obese REG3A-TG mice showed a decrease in basal fasting C-peptide compared with the obese WT mice (Fig. 3c–e). In response to glucose, plasma insulin and C-peptide were elevated in obese WT mice compared with basal levels before gavage. In contrast, obese REG3A-TG mice showed lower stimulated insulin and C-peptide (Fig. 3d, e). Thus, REG3A-TG obese mice require 50% less insulin than WT obese mice to achieve similar glucose homeostasis, suggesting that REG3A may protect against the deleterious effects of a high-fat diet on insulin sensitivity and insulin secretion. When the HOMA-IR was calculated, there was no significant difference between the two mouse lines, although a small trend of decrease was noted in the REG3A group (Fig. 3f). In an insulin tolerance test, REG3A-TG obese mice exhibited a greater hypoglycemic response to exogenous insulin than WT obese mice, consistent with the increased insulin sensitivity they exhibit when fed a standard diet (Fig. 3g). Preserved glucose homeostasis in REG3A-TG obese mice occurs without any change in body weight (Fig. 3b). Furthermore, comparable levels of Akt phosphorylation (Ser473) in the liver, tibialis anterior, soleus, and WAT of obese REG3A-TG and control mice seem to exclude that the mechanism underlying the enhanced metabolic phenotype of obese REG3A-TG mice is an increase in Akt-dependent insulin signaling (Supplementary Figure 5a-d). The morphology of the pancreatic beta islets appeared normal in the transgenic mice. We detected no difference between islets from TG-REG3A and WT obese mice after immunostaining for insulin, suggesting that the lower insulin secretion under basal conditions and during glucose tolerance test in high-fat-fed REG3A-TG mice would not be related to altered pancreatic beta cell function (Fig. 3h). These results establish that REG3A overexpression is associated with reduced insulin secretion in response to hyperglycemic challenge and improved insulin sensitivity under high-fat feeding.

**REG3A reduces hyperglycemia and insulin resistance in ob/ob mice.** We next hypothesized that the administration of a full-length recombinant REG3A protein (rcREG3A, ALF5755) would

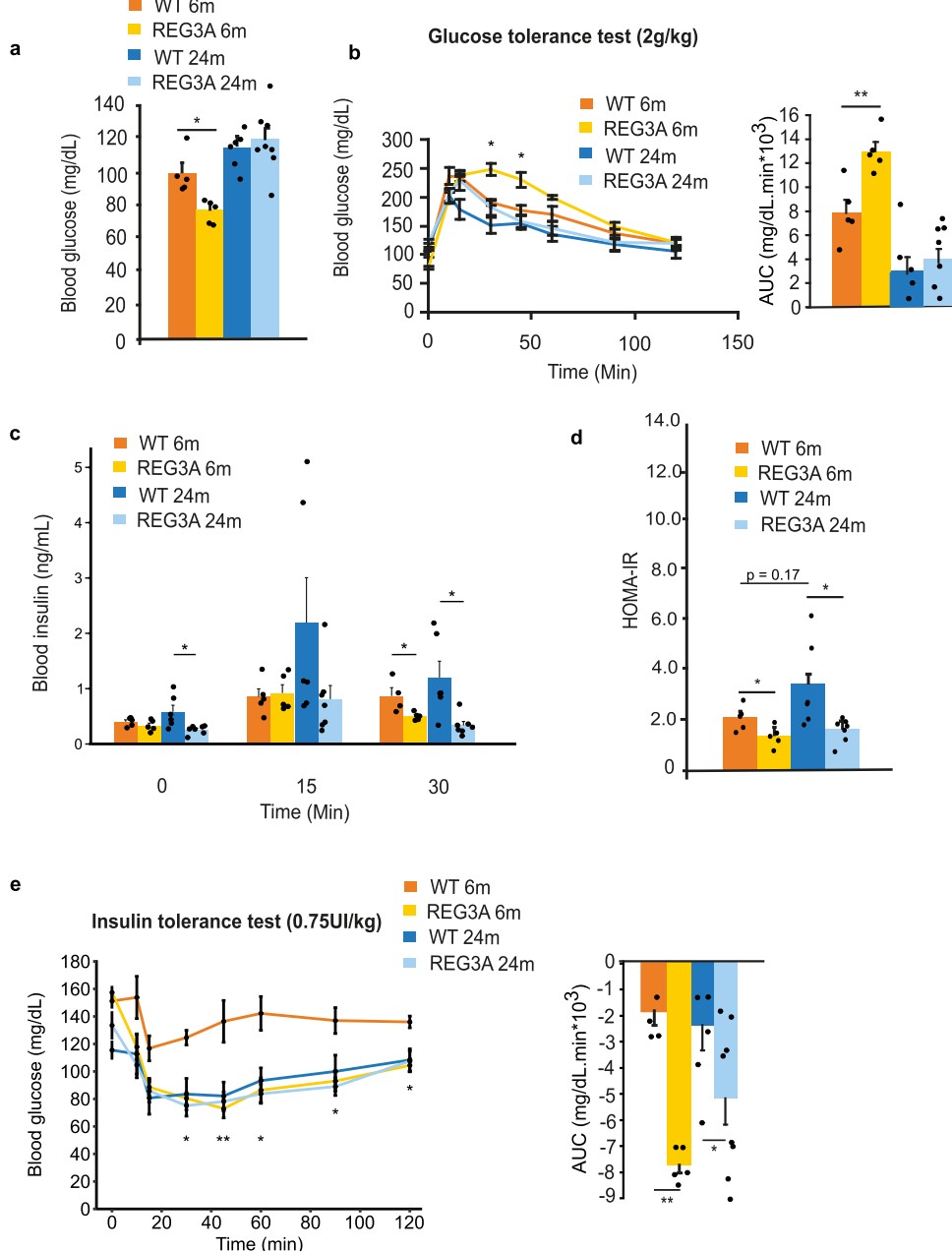

**Fig. 2 REG3A transgenic mice fed a standard diet are hypersensitive to insulin.** Glucose homeostasis was assessed in 6- and 24-month-old REG3A transgenic mice (REG3A) and wild-type (WT) mice ($n = 8$ for each group, same as Fig. 1) fed ad libitum with standard chow. **a** Fasting blood glucose. **b** Blood glucose curves under oral glucose tolerance tests OGTT (baseline at time $t = 0$ right before glucose gavage). Right: area under the curve of OGTT. **c** Insulin levels during same OGTT as Fig. 2b. **d** Insulin sensitivity estimated by HOMA-IR (homeostatic model assessment for insulin resistance). **e** Blood glucose curves under insulin tolerance tests ITT (baseline at time $t = 0$ right before insulin injection). Right: area under the curve of ITT. Asterisks indicate significant differences between WT and REG3A at 6 months of age. Data are averages ± SEM. $*p < 0.05$; $**p < 0.01$ by Anova (**a**, right **b**, **c**, **d**, right **e**) or repeated measures Anova (left **b**, left **e**) both followed by a post hoc test. NS or no statistical indication, no significance.

improve metabolic homeostasis and insulin resistance and tested that hypothesis first in the ob/ob mice, a congenital leptin-deficient model of severe obesity, hyperinsulinemia and insulin resistance. We have previously shown that ALF5755 is a biologically active protein that binds to oligo- and polysaccharides, but not to monosaccharides[17], and exhibits cytoprotective and anti-oxidant activity promoting tissue regeneration in vivo[20,23,51,52]. Twelve-week-old ob/ob mice were infused with a daily dose of 9 µg or 43 µg of rcREG3A (or an equivalent volume of buffer) using an osmotic pump-based delivery system for 28 days (Fig. 4a). This resulted in serum REG3A levels of 100 ± 28 and

250 ± 44 ng/mL for 9 µg or 43 µg of administered rcREG3A per day, respectively (Fig. 4b). Both quantities of infused rcREG3A were chosen to fit within the range of circulating REG3A levels as found in vivo in TG-REG3A mice. Body weight and fat mass gains were similar between ob/ob mice infused with vehicle or rcREG3A (Fig. 4c, d). However, significant differences were observed in several blood metabolic parameters, including a reduction in fasting concentrations of nonesterified (free) cholesterol, triglycerides, neutral lipids, and glucose by 29%, 74%, 23%, and 30%, respectively (Fig. 4e, f). There was no difference in hepatic lipid concentration and liver steatosis score

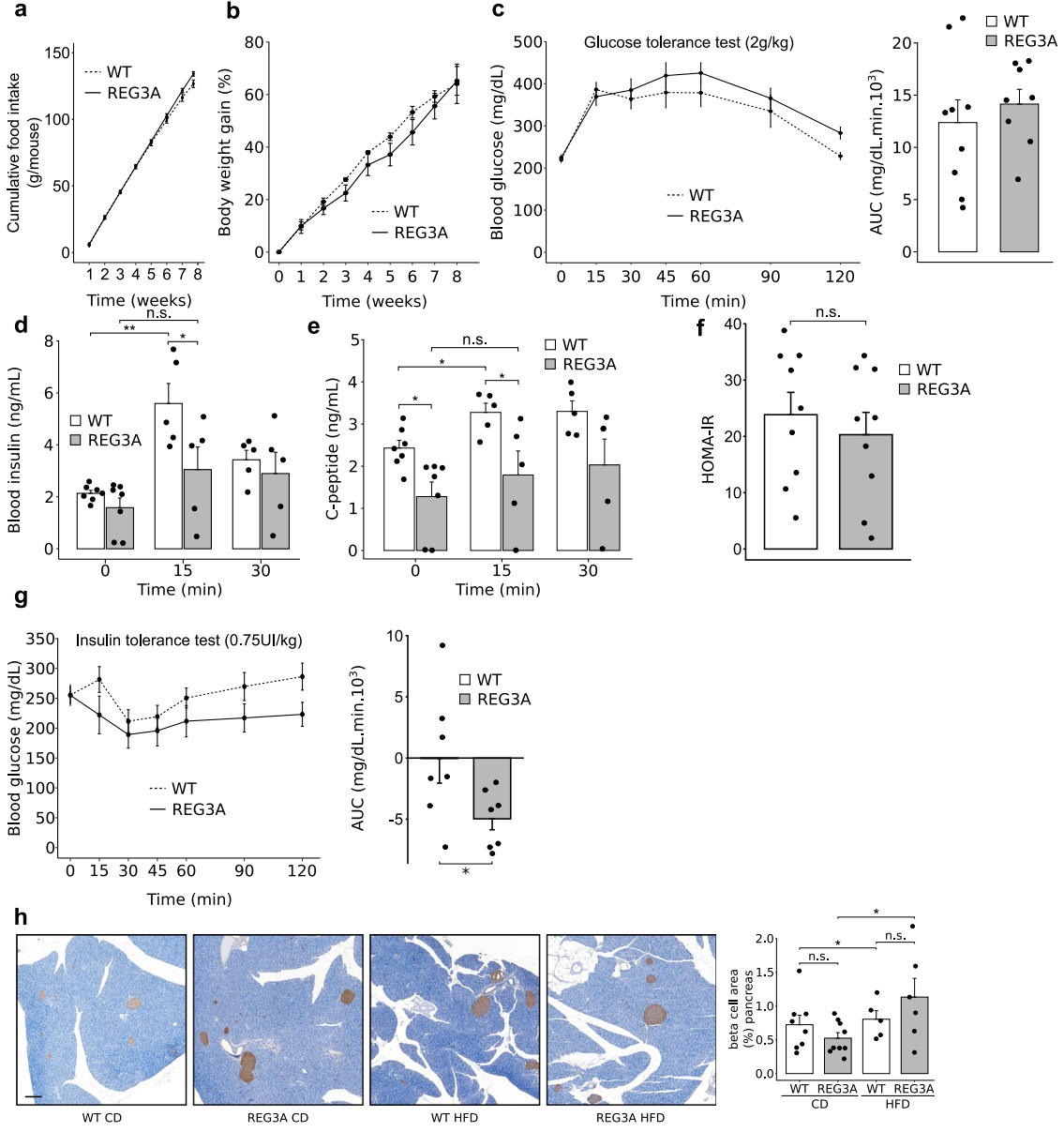

**Fig. 3 Improved insulin sensitivity and reduced insulin secretion in obese REG3A transgenic mice fed a high fat diet.** Four-month-old REG3A transgenic mice (REG3A) and wild-type (WT) mice ($n = 8$ REG3A, $n = 9$ WT, new cohorts different from Figs. 1 and 2) were fed a high-fat, butter-enriched diet (5505 kcal/kg) for 8 weeks and then analyzed. **a** cumulative food intake. **b** Body weight gain. **c** Blood glucose curves under oral glucose tolerance tests OGTT. Right: area under the curve of OGTT. **d**–**f** Blood tests (**d**) Blood insulin, **e** C-peptide concentrations and **f** Insulin resistance indices (HOMA-IR). **g** Blood glucose curves under insulin tolerance tests ITT. Right: area under the curve of ITT assays. ($n = 7$ per group). **h** Representative images of insulin staining on pancreas sections from REG3A transgenic and WT mice fed standard (CD) or a high-fat (HFD) diet, and quantification of insulin positive areas. Scale bar: 200 μm. Data are averages ± SEM. ∗$p < 0.05$ by Anova test (**d**, **e**, **h**), one-way repeated measures Anova (**a**, **b**, left **c**, left **g**) both followed by post-hoc test, Wilcoxon rank sum test (right **c**, right **g**). NS or no statistical indication, no significance.

(Supplementary Fig. 6a, b). For mice injected with 9 μg of rcREG3A, glucose concentration decreased from $283 ± 24$ mg/dL on day 1 to $193 ± 6$ mg/dL 11 days later and, for mice injected with 43 μg of rcREG3A, from $316 ± 32$ mg/dL to $234 ± 19$ mg/dL. Glucose levels remained stable afterwards for mice injected with both rcREG3A doses (Fig. 4f). In an insulin tolerance test, the insulin response was significantly improved in ob/ob mice given either dose of rcREG3A compared with the vehicle group, demonstrating that administration of rcREG3A over a 4-week period greatly improves systemic insulin sensitivity (Fig. 4g, h). With the same levels of glucose clearance whether or not the mice were treated with rcREG3A, there was a significant decrease in circulating insulin levels at 15 min and a trend at 30 min after a

glucose tolerance test, suggesting better control of insulin resistance in those treated with rcREG3A (Fig. 4i–k). This was illustrated by a stabilization of HOMA-IR values in rcREG3A-treated mice, and a further increase in the vehicle group, indicating attenuation of insulin resistance by rcREG3A (Fig. 4l). There was no significant difference in insulin sensitivity (HOMA-IR) of ob/ob mice receiving a dose of 9 μg or 43 μg per day at D11 or D25 after rcREG3A infusion (D11: $P = 0.39$; D25: $P = 0.063$), so we continued with the larger dose of rcREG3A. Overall, ob/ob mice treated with rcREG3A showed an increase in the hypoglycemic effect of exogenous insulin as well as a decrease in circulating insulin levels under glucose-stimulated conditions. These results conclusively demonstrate that insulin resistance was significantly

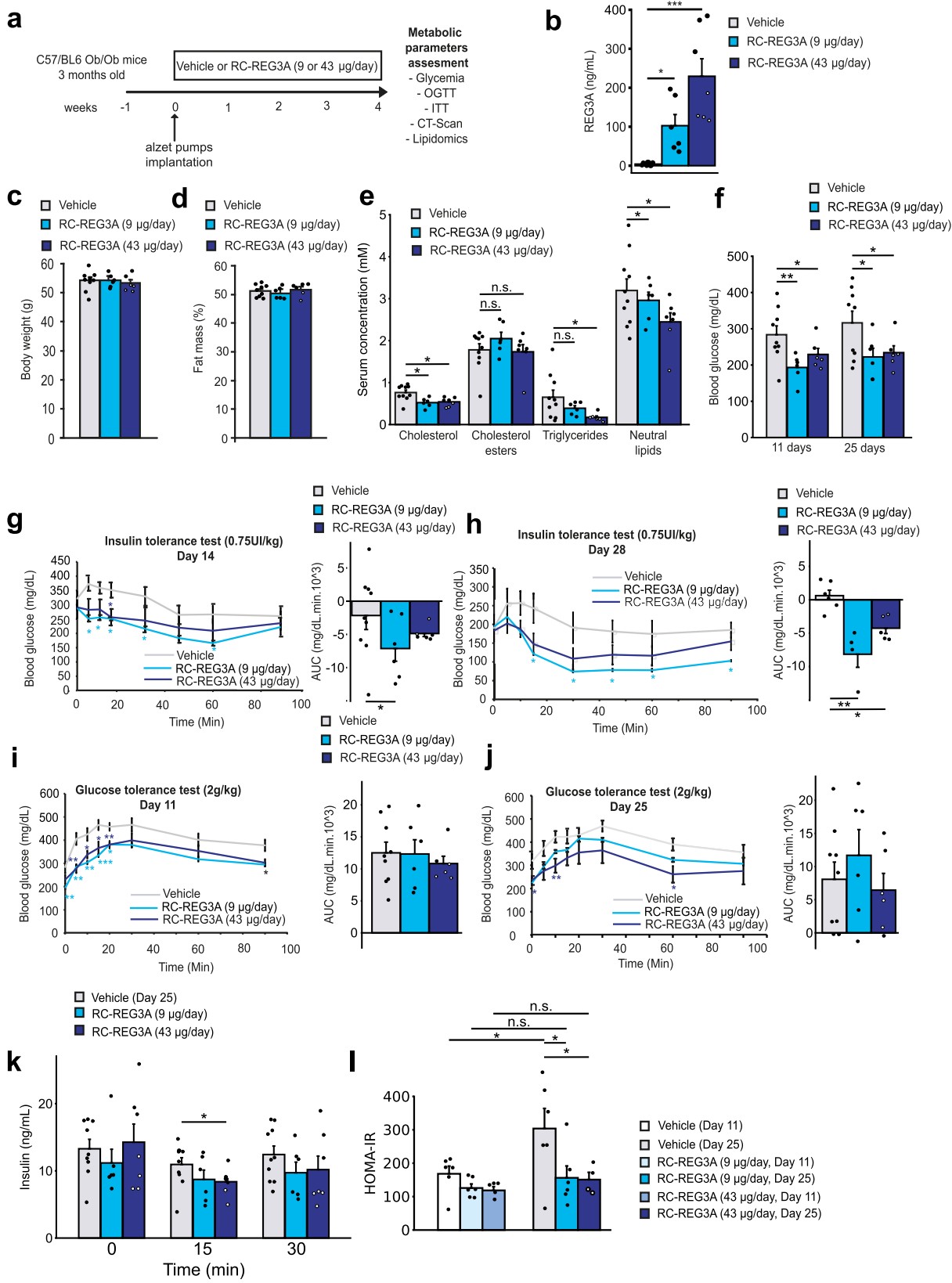

attenuated by chronic REG3A treatment in both diet-induced obesity and that resulting from the ob/ob model.

**REG3A reduces glucose tolerance and dyslipidemia in pre-diabetic mice.** To examine whether the insulin-sensitizing action

of rcREG3A is effective in non-genetically obese mouse models with insulin resistance, we tested the effect of rcREG3A on glucose metabolism in C57BL/6 WT mice with HFD-induced pre-diabetes (Fig. 5a). After 10 weeks of HFD feeding (HF235; 4655 kcal/kg; 37.5% carbohydrate, 27.5% fat, 17% protein), WT mice given 43 µg per day of rcREG3A for 28 days showed

**Fig. 4 Ob/ob mice given a recombinant REG3A protein are less hyperglycemic and insulin resistant.** Three-month-old Ob/Ob mice were given a daily subcutaneous dose of 9 µg or 43 µg of a recombinant human REG3A protein (rcREG3A) for 28 days ($n = 10$ per group). Control ob/ob mice received an equivalent volume of buffer (vehicle). Glucose homeostasis was assessed on days 11 and 25 after starting rcREG3A or buffer treatment. **a** Diagram of the experimental protocol. OGTT: oral glucose tolerance test. ITT: insulin tolerance test. **b–e** Measurement of indicated markers on day 29 (end of study). **b** Serum concentrations of REG3A. **c** Body weight. **d** Adipose tissue content. **e** Serum levels of lipids. **f** Fasting blood glucose on days 11 and 25 after onset of treatment. **g, h** Blood glucose curves under insulin tolerance tests on (**g**) day 14 and (**h**) day 28. **I, j** Blood glucose curves under glucose tolerance tests on (**i**) day 11 and (**j**) day 25. **k** Blood insulin concentrations and **l** insulin resistance indices (HOMA-IR) after oral glucose intake on day 25. Data are averages ± SEM. $*p < 0.05$; $**p < 0.01$; $***p < 0.005$ by Anova test (**b–f**, **k**, **l**, right panels of **g**, **h**, **i** and **j**) or one-way repeated measures Anova (left panels of **g**, **h**, **i** and **j**) both followed by post-hoc test. NS or no statistical indication, no significance.

---

improvement in metabolic syndrome and associated traits (changes in lipid profile, leptin, glucose tolerance) not seen in mice given the vehicle (Fig. 5). Weight gain and adiposity were equivalent in mice treated with rcREG3A or vehicle (Fig. 5b, c). In contrast, significant reductions in blood lipid and leptin levels of about 30% were observed in mice given rcREG3A which is consistent with the results obtained in ob/ob mice, as well as a trend toward reduced hepatic steatosis (Fig. 5d–f). While the HF235 diet did not alter fasting blood glucose compared with the chow diet, rcREG3A-treated mice had lower blood glucose levels than control mice (Fig. 5g). In response to glucose challenge, rcREG3A significantly improved glucose tolerance and reduced glucose-stimulated insulin secretion by −56% at 15 min and −45% at 30 min (Fig. 5h, i). This differs from the impaired glucose tolerance and threefold increase in insulin secretion required to adapt to the glucose challenge in HF235-fed mice receiving a vehicle (Fig. 5i). We detected no difference between islets from mice fed HF235 and treated, or not, with rcREG3A after immunostaining for insulin (Supplementary Fig. 7). Such improvements in glucose homeostasis in rcREG3A-treated obese mice was not accompanied by Akt phosphorylation on Ser473 in the peripheral tissues analyzed (tibialis anterior, liver) (Supplementary Fig. 8). These observations indicate that rcREG3A reduces the metabolic consequences of an unbalanced fatty diet, including the correction of dyslipidemia, by decreasing insulin secretion in response to glucose at a prediabetic stage. This suggests that REG3A may enhance insulin action through an Akt-independent mechanism and thus contribute to curb the development of insulin resistance.

**REG3A increases glucose uptake in glycolytic skeletal muscles, but not in oxidative skeletal muscles and WAT.** To provide insights into the mechanisms that contribute to the control of glucose metabolism by rcREG3A, and to identify the tissue(s) where insulin action is enhanced, we performed hyperinsulinemic-euglycemic clamp (3mU/minKg) studies in WT mice with HF235-induced prediabetes or fed a chow diet and given rcREG3A or vehicle for 28 days. We showed that the rcREG3A-treated group had a glucose turnover rate comparable to that of the vehicle group in both the standard and high-fat diets. The rate of glucose infusion to maintain euglycemia was similar in mice given rcREG3A or a vehicle in response to HF235, indicating equivalent whole-body insulin sensitivity in both setups. Endogenous glucose production was also the same in all groups indicating no particular effect of rcREG3A on this parameter (Fig. 5k). Labeled 2 deoxy-glucose was injected at the end of the clamp in order to measure glucose uptake in skeletal muscles either oxidative (soleus) or glycolytic (tibialis anterior) as well as fat pad (Fig. 5l). As shown in Fig. 5, the tibialis anterior of HF235 fed mice that received rcREG3A showed a significant increase in insulin-stimulated glucose uptake compared with vehicle-treated mice, whereas the latter developed skeletal muscle insulin resistance during HF235. In contrast, adipose tissue and soleus showed glucose uptake comparable to that of control mice

(Fig. 5l). This suggests that REG3A regulates glucose homeostasis and insulin sensitivity in a tissue-specific manner, this phenomenon is presumably related to the fast muscle glucose clearance in the targeted skeletal fiber muscles.

**REG3A attenuates oxidative protein damage and activates AMPK in skeletal muscle.** We, and others, have previously demonstrated that REG3A is a coupling agent between innate immunity and organ homeostasis through multiple functionalities, such as antimicrobial, antioxidant, survival factor, and regulator of inflammatory response in various damaged tissues, where REG3A promotes tissue repair and regeneration[1,2,17,20,21,23,53]. Serum levels of tumor necrosis factor α (TNF-α), IL-6, and interferon-γ were measured in mice over-expressing REG3A or infused with rcREG3A to define their relationship with REG3A-triggered metabolic changes. Elevations in cytokine levels were not substantial in response to HFD feeding or leptin deficiency, and did not differ significantly in the presence of the transgene or recombinant REG3A (except for IL6 in mice fed HF235 and given rcREG3A) (Supplementary Fig. 9). Because REG3A is a ROS scavenger, we wondered whether REG3A's ability to counteract oxidative damage, particularly in muscle, could help improve glucose homeostasis, which is chronically impaired by high-fat diets and obesity. Protein carbonylated levels were measured in tissues and muscle cells by derivatization of carbonyl groups with 2,4-dinitrophenylhydrazine (DNPH) or biotin-hydrazine, followed by immunodetection with a DNPH-specific antibody (Fig. 6b) or by affinity capture (Fig. 6j). We found higher levels of protein carbonylation in the tibialis muscle of ob/ob mice than in that of HF235-fed mice (the latter showing low levels of oxidative damage comparable to that seen during a standard diet), an effect significantly attenuated by rcREG3A in ob/ob mice (Fig. 6a). Similarly, quantification of protein carbonylation in differentiated C2C12 myotubes yielded 4-fold higher values under glucose oxidase (GO)-induced oxidative stress than in the absence of GO. The accumulation of oxidized proteins in C2C12 cells was significantly reduced by rcREG3A, indicating antioxidant activity of REG3A in skeletal muscle cells and tissue (Fig. 6b). To determine whether reduction of muscle oxidative damage by REG3A improves glucose control, we tested basal and insulin-induced phosphorylation of Akt in oxidized C2C12 cells. GO completely inhibited insulin-induced phosphorylation of Akt on Ser473 in C2C12 cells and rcREG3A did not reverse the oxidative stress-induced impairment of insulin action (Fig. 6c). We turned to AMP-activated protein kinase (AMPK), another key regulator of metabolism, and showed that the level of phosphorylation of AMPK (i.e., the active form of AMPK) was higher in rcREG3A-treated C2C12 cells than in untreated cells under baseline and oxidative stress conditions (Fig. 6 d). The effect of rcREG3A on AMPK activation was dose-dependent in C2C12 myotubes and a trend was observed in primary myocytes (Fig. 6 e, f). Importantly, we found a positive association between rcREG3A and AMPK phosphorylation, and a neutral effect on Akt activation in the

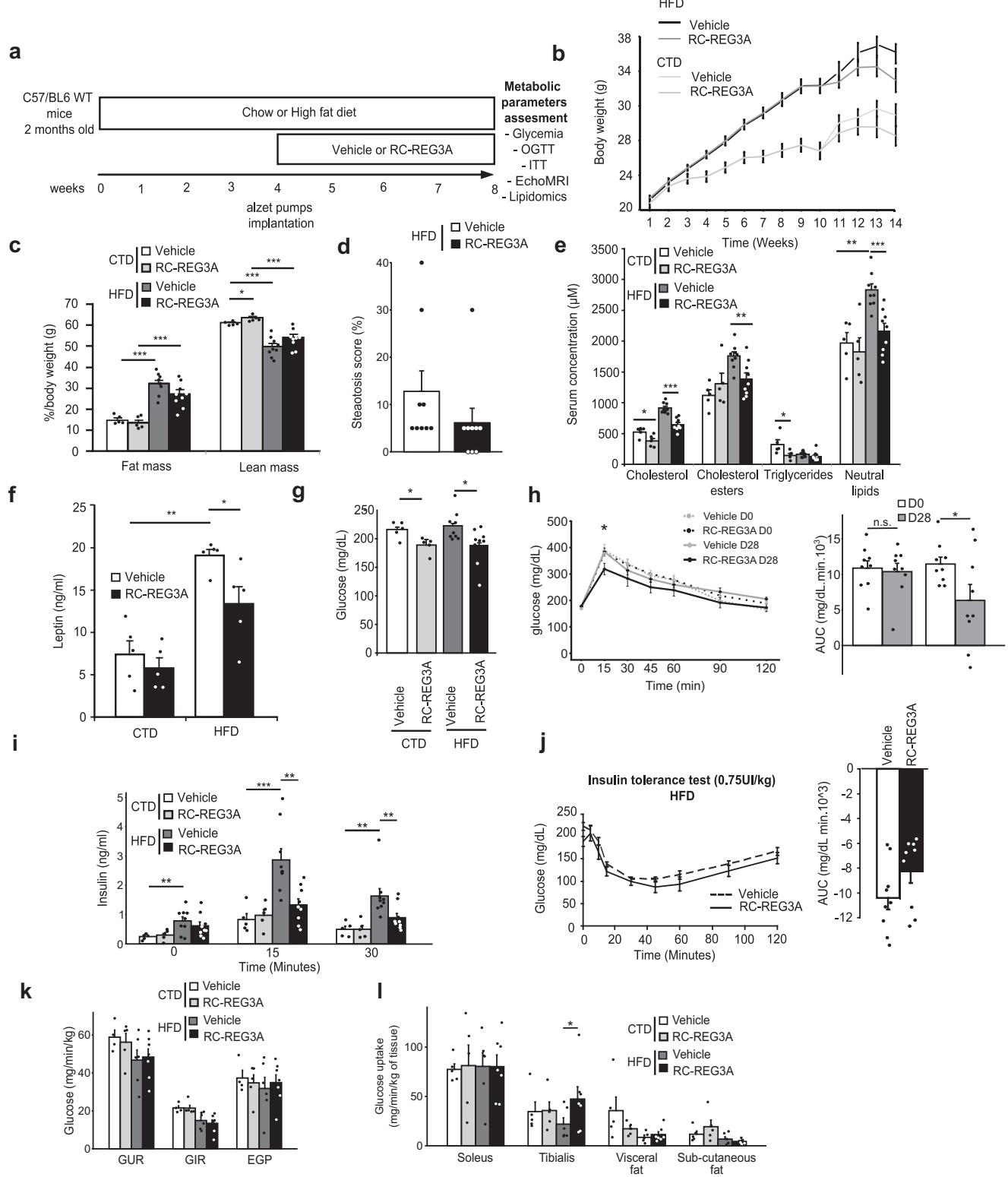

tibialis of HF235-fed prediabetic and ob/ob mice (Fig. 6 g, h). Next, we investigated how the increase in AMPK activity was enhanced by REG3A and the specific contributions of glycoprotein 130 (gp130) signaling. Reg3 family members are linked to components of several signaling pathways, depending on the tissue, cell, and disease type. These include the gp130 receptor complex, which is one of the triggers of muscle AMPK activation through IL6 and ciliary neurotrophic factor (CNTF) signaling[3,54].

On the other hand, IL-6 family cytokines (CNTF, LIF, IL6) that use the gp130 pathway are positive signals triggering changes in gene expression of Reg3b, the murine homolog of REG3A, in many cell types[4,5,55–57]. Finally, the gp130-JAK2-STAT3 cascade is reported to mediate the effects of Reg3b[6–8,58]. We found that REG3A-mediated AMPK activation is gp130 dependent, as gp130 silencing significantly suppressed AMPK activation in C2C12 cells stimulated with rcREG3A (Fig. 6i). We then

**Fig. 5 Obese prediabetic mice given a recombinant REG3A protein showed improved glucose homeostasis and increased glucose uptake by skeletal muscle.** After 10 weeks of HFD (4655 kcal/kg), WT mice received 43 µg per day of a recombinant REG3A protein (rcREG3A) subcutaneously for 28 days (n = 9). Mice on a standard diet (chow, CTD) also received rcREG3A for the same period (n = 5). HFD-fed (n = 10) and CTD-fed control mice (n = 5) received an equivalent volume of buffer (vehicle). **a** Diagram of the experimental protocol. OGTT: oral glucose tolerance test. ITT: insulin tolerance test. **b–k** Measurement of indicated markers at the end of rcREG3A or buffer treatment. **b** Body weight over time. **c** Body fat and lean mass. **d** Histological score of liver steatosis. **e** Serum levels of lipids. **f** Serum levels of leptin. **g** Fasting blood glucose. **h** Blood glucose curves under oral glucose tolerance tests OGTT. Right: area under the curve of OGTT. **i** blood insulin concentrations during glucose tolerance test. **j** Blood glucose curves during insulin tolerance test. Right: area under the curve of ITT. **k** Hyperinsulinemic-euglycemic clamp in mice fed a chow (CTD, n = 10) or a high-fat (HFD, n = 14) diet and treated subcutaneously with 43 µg per day of a recombinant REG3A protein (rcREG3A) or an equivalent volume of buffer for 28 days. GUR: Whole-body glucose utilization rate. GIR: glucose infusion rate; EGP: endogenous glucose production. **l** Tissue sensitivity to insulin estimated by the quantification of glucose uptake in the indicated tissues. Data are averages ± SEM. $*p < 0.05$; $**p < 0.01$; $***p < 0.005$ by Anova test (**c**, **e**, **f**, **g**, right **h**, **i**, **k**, **l**) or one-way repeated measures Anova (left **h**, left **j**) both followed by post-hoc test, Wilcoxon rank sum test (right **j**). NS or no statistical indication, no significance.

hypothesized that the carbonyl levels of gp130 and the insulin receptor might be affected by REG3A given the significant reduction in muscle protein carbonylation associated with REG3A in vivo and in vitro. We used affinity capture and streptavidin-linked beads, followed by immunodetection with antibodies specific for gp130 or the insulin receptor, to assess the amount of these oxidized membrane proteins. Addition of rcREG3A to GO-exposed C2C12 cell culture significantly reduces carbonylation of gp130 but not of the insulin receptor (Fig. 6j), suggesting that REG3A, by protecting gp130 from oxidative damage, releases the gp130-dependent AMPK activation pathway.

**Changes in REG3A expression in metabolic organs of obese and diabetic patients**. We explored the human relevance of our findings by investigating whether altered glucose metabolism in obese/diabetic patients could be correlated with aberrant REG3A expression. We analyzed endogenous REG3A expression using publicly available large-scale microarray datasets from human tissues of individuals with different metabolic phenotypes. As expected, REG3A mRNA expression is mainly restricted to the intestine and pancreas, and low or nearly absent in liver, adipose tissue, and skeletal muscle under physiological conditions (Supplementary Fig. 10a). We selected metabolic disorder datasets deposited in the Gene Expression Omnibus (GEO)[59–63] and examined REG3A gene expression in major metabolic tissues (i.e. liver, pancreas, WAT, vastus lateralis muscle). Significant changes were observed in WAT and mostly muscle between insulin-sensitive and diabetic states, but not in liver and pancreas. A significant decrease in REG3A expression in muscle was observed in the insulin-resistant and diabetic conditions compared with the insulin-sensitive condition, suggesting that REG3A in skeletal muscle may be involved in glucose-insulin homeostasis (Fig. 7a). The public availability of skeletal muscle transcriptomic data obtained over time after metabolic surgery (2, 12, 24, 52 weeks; GSE 135066) allowed us to study any changes in REG3A gene expression. While REG3A expression was low in the muscle before surgery, it increased significantly upon resolution of the disease before returning to basal levels at 52 weeks (Fig. 7b). The temporal variation in REG3A coincided with transient changes in the expression of many genes, including those related to insulin signaling, AMPK signaling and fatty oxidation, and mitochondrial function in skeletal muscle (Fig. 7c–e). The level and timing of REG3A induction were not shared with the vast majority of molecules in the antimicrobial peptide and protein class to which REG3A belongs (Fig. S10b–d). Only REG3 gamma (REG3G) and Defensin Alpha 5 (DEFA5) showed similar expression dynamics to REG3A during the disease resolution period, whereas the expression of almost the entire set of AMPs analyzed was negatively modulated (n = 25) or not modulated at all (n = 56) after

bariatric surgery. This in silico analysis suggests a strong link between improved muscle insulin sensitivity and REG3 proteins.

**Discussion**
Mediators of the immune response help control energy metabolism, and disruption of one or more of these inter-organ signals can have dramatic effects on the pathogenesis of insulin homeostasis in obesity and type 2 diabetes mellitus[64]. Chronic systemic and local low-grade inflammation and oxidative stress mediators are important pathogenic factors in this setting, suggesting that control of inflammatory/oxidative defects may help resolve insulin resistance and diabetes. Indeed, evidence in human and animal models indicate that metabolic oxidative stress is a trigger in the pathogenesis of insulin resistance, diabetes and its complications. In diabetes and obesity, glucolipotoxicity among other processes prompts an overproduction of ROS overwhelming the scavenging capabilities of metabolic tissues. Increase in non-scavenged intermediates leads to oxidative damage. Protein carbonylation that irreversibly impairs protein structure and function is an example of such damage. Carbonylation of insulin receptor substrates 1 and 2 by ROS contributes to a loss of insulin responsiveness, resulting in compensatory insulin hypersecretion. High ROS concentrations and subsequent overactivation of ROS-responsive pathways enhance oxidative damage and lower insulin efficacy[65–67]. In addition, elevated ROS production caused by mono-oleoyl-glycerol stimulates basal insulin secretion in pancreatic ß cells[68]. In turn, the accumulation of oxidized, misfolded and aggregated proteins exacerbates oxidative stress levels, insulin resistance and diabetes.

We and others have shown that acute-phase REG3 protein expression increases after tissue injury. The resulting elevated blood concentration of REG3A has been proposed as a possible prognostic biomarker for many human acute and chronic diseases, including pancreatitis, inflammatory bowel disease, graft versus host disease, and ischemic stroke[69–72]. A growing body of research has shown that overexpression of REG3A promotes the resolution of inflammation and tissue repair in different experimental settings by controlling key repair mechanisms such as wound repair, immune cell recruitment, and proliferation. In the liver, REG3A pairs up with extracellular matrix components of inflamed tissue, and the resulting high local concentration of REG3A in the damaged area makes it a particular effective antioxidant that contributes to better-controlled regeneration after tissue injury[51]. In this study, we establish links between oxidative stress regulation and metabolic regulation though REG3A. We report here that increase of REG3A, both as a transgene and as a recombinant protein, controls elevated basal insulin levels during aging, high-fat feeding, and in obese ob/ob mice and prevents glucose-induced hyperinsulinemia. REG3A reduces hyperglycemia and dyslipidemia in prediabetes and type 2 diabetes in obese rodents. A basis for the increased whole-body

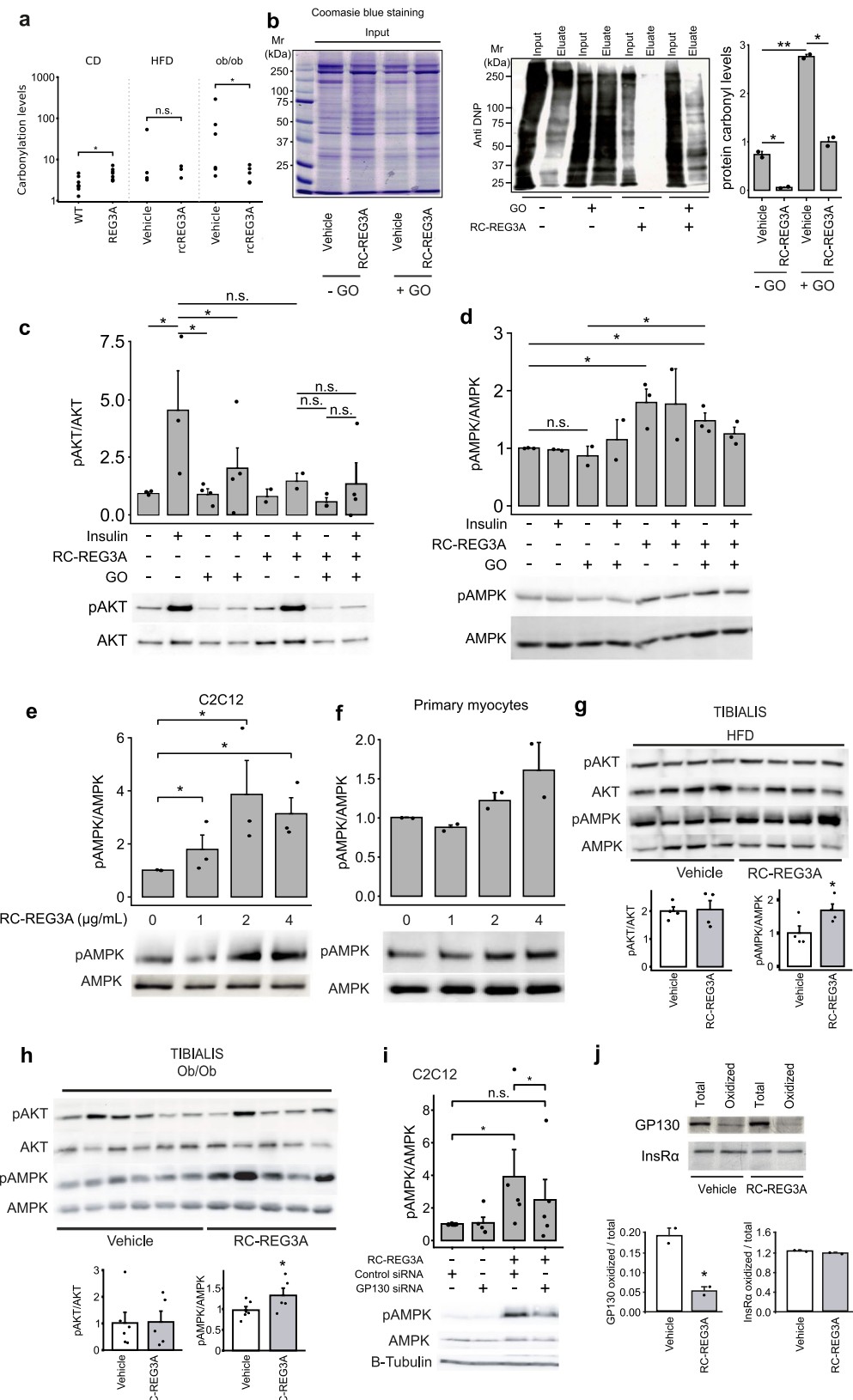

insulin sensitivity was found to be the effectiveness of REG3A in reducing oxidative stress and activating AMPK signaling in skeletal muscle. This phenomenon is not associated with changes in body fat levels for the recombinant REG3A protein. In contrast, in REG3A transgenic mice, improvement of metabolic phenotype was associated with changes in primary adiposity. Because ectopic lipid accumulation is clearly associated with insulin-resistant states, proper maintenance of insulin homeostasis in aging REG3A mice may be a by-product of the marked reduction in body fat in these mice. It is conceivable that the observed effect of REG3A protein on glucose uptake in glycolytic skeletal muscle and not in fat helps counteract excessive triglyceride storage in

**Fig. 6 The antioxidant rcREG3A increases AMPK activation in the skeletal muscle. a** Quantification of normalized protein carbonylation levels in tibialis muscle of wild-type (WT) and REG3A transgenic (REG3A) mice fed a standard diet (CD, $n = 8$), mice fed a high-fat diet (HFD, $n = 4$), and ob/ob mice receiving 43 μg per day of recombinant REG3A protein (rcREG3A, $n = 4$–6) for 28 days. Control mice fed the HFD diet and ob/ob received an equivalent volume of buffer (vehicle). **b** Left: Coomassie blue staining: protein loading control. Middle: Anti-DNP immunoblots displaying total carbonylated proteins from C2C12 myoblasts in the absence (−) or presence (+) of recombinant REG3A protein (rcREG3A) and exposed or not to glucose oxidase (GO)-induced oxidative stress. Eluate: carbonylated protein enrichment using biotin tagging and avidin affinity chromatography. Right: densitometry quantification of immunoblots performed on affinity-enriched carbonylated proteins (Eluate lane). **c, d** Representative immunoblots using the indicated metabolic markers in extracts of C2C12 myoblasts treated or not with insulin and recombinant REG3A protein (rcREG3A) without or with oxidative stress (GO). AKT and AMPK immunoblots are shown as protein loading controls. Below: densitometry quantification of the immunoblots. Three independent experiments. **e–f** Anti-phospho-AMPK (pAMPK) immunoblotting in mouse C2C12 cells (**e**) and primary myocytes (**f**) for different doses of rcREG3A. Bar chart: densitometric analysis. **g, h** Representative immunoblots for the indicated markers in muscle extracts (**g**) in the high fat diet (HFD)-fed WT mice of Fig. 5 and (**h**) in the ob/ob mice of Fig. 4 treated with rcREG3A or buffer (vehicle). Each lane represents 1 mouse. Bar chart: Densitometric analysis. **i** Representative immunoblots for the indicated markers in C2C12 myoblasts with or without rcREG3A treatment and with downregulated gp130 expression (gp130 siRNA) or not (Control siRNA). Bar chart: densitometric analysis. Two independent experiments. **j** Total and carbonylated levels (oxidized) of GP130 and insulin receptor (InsR alpha) in C2C12 myoblasts under oxidative stress conditions treated with a recombinant REG3A protein (rcREG3A) or buffer (vehicle). Two independent experiments. Means of three independent experiments ± SEM. ∗$p < 0.05$; ∗∗$p < 0.01$ by Anova test followed by post-hoc test (**b–f**, **i**) or Student's $t$ test (**a**, **g**, **h**, **j**). NS or no statistical indication, no significance.

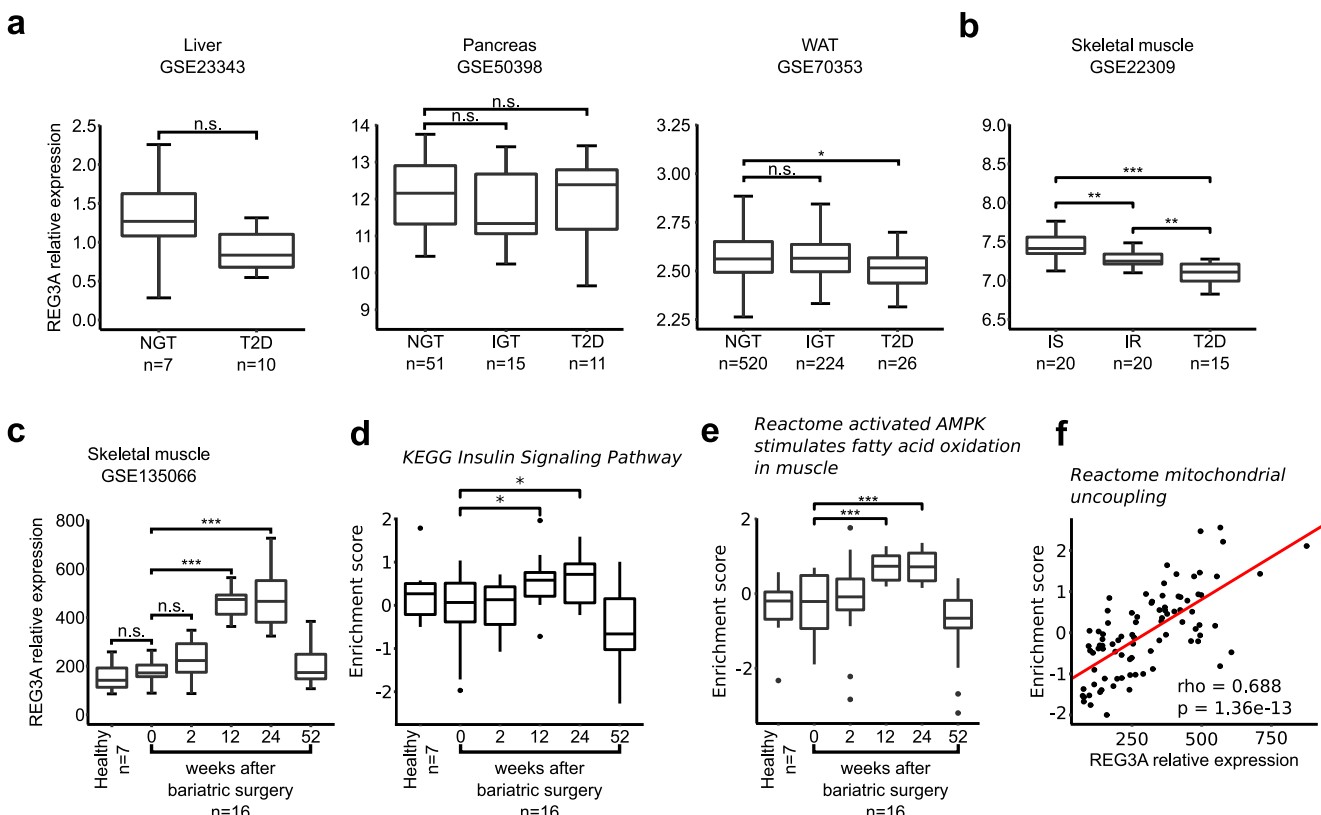

**Fig. 7 Endogenous upregulation of REG3A in skeletal muscle is linked to improved metabolism after bariatric surgery in obese individuals. a, b** REG3A mRNA expression level assessed from transcriptomes of indicated tissues in patients with metabolic disease. **a** NGT: normal glucose tolerance (Hb1Ac less than 6%); IGT: impaired glucose tolerance (HbA1c between 6 and 6.5%); T2D: type 2 diabetes (HbA1c greater than 6.5%); **b** IS: insulin-sensitive; IR insulin-resistant. IS et IR were defined by measuring the glucose disposal rate during an hyperinsulinemic euglycemic clamp in Wu et al. Endocrinology, et al. 2007. **c–f** Molecular changes over time in the vastus lateralis skeletal muscle of 16 patients who underwent bariatric surgery. **c** REG3A mRNA expression. **d** Gene enrichment of "Insulin signaling pathway" in KEGG. **e** Gene enrichment of "Activated AMPK stimulates fatty acid oxidation in muscle" in Reactome. **f** Spearman correlation between REG3A expression and enrichment score of the "mitochondrial uncoupling pathway" in Reactome. ∗$p < 0.05$; ∗∗$p < 0.01$; ∗∗∗$p < 0.005$ by Anova test followed by post-hoc test. NS or no statistical indication, no significance.

adipocytes and explains the decrease in fat mass and increase in lean mass in REG3A transgenic mice over time. In addition, it should be noted that glycolytic muscles (tibialis anterior) are more vulnerable to sarcopenia than oxidative muscles (soleus) possibly related to a higher generation of ROS in glycolytic muscles during aging[73–75]. The attenuation of muscle oxidative damage by REG3A stems from its scavenging activity against two deleterious ROS, namely, the superoxide anion and hydroxyl radical, both of which are known to be abundantly produced during chronic metabolic disease[76,77]. Similar ROS scavenging activity involved in the protection of murine hematopoietic stem cells has been reported for other lectins[78]. Because insulin resistance in skeletal muscle is the major disruption of insulin action in type 2 diabetes[79,80], increasing muscle action of REG3A may

contribute to the decrease of insulin resistance in diabetes and obesity. The decrease in membrane carbonylated protein content by REG3A in skeletal muscle strongly suggests that the significant changes in insulin sensitivity in obese/diabetic mice administered REG3A are related to the antioxidant activity of REG3A in skeletal muscle. Such a REG3A-mediated mechanism of action most likely promotes induction of gp130-AMPK signaling and thus bypasses the typical impairment of insulin signaling, leading to minimization of the amount of insulin needed to control blood glucose and reduce hyperinsulinemia. Similar effects of reduced insulin hypersecretion and dyslipidemia are observed with REG3A transgenic mice during aging and obesity after feeding with a high-fat diet. While the recombinant REG3A protein bypasses the AKT-dependent insulin pathway by playing on the AMPK pathway in muscle, we found indications that the REG3A transgene and consequently the resulting overexpression of circulating and intrahepatocyte REG3A may play on the enhancement of alternative Akt and AMPK signaling pathways in the liver (Supplementary Fig. 11). These results suggest that REG3A contribute to a strong control of glucose homeostasis while significantly reducing hyperinsulinemia and improving insulin sensitivity.

Together, this study shows that REG3A significantly reduces oxidative protein damage in skeletal muscle and improves glucose homeostasis and insulin sensitivity by activating AMPK in obese/diabetic mice. Through this mechanism, REG3A could potentially preserve the fitness of metabolic organs such as adipose tissue, muscle, liver and pancreas. Our results suggest that a therapeutic approach based on REG3A could break the vicious circle between hyperinsulinemia and altered signaling by oxidative stress, which, if not controlled, contributes to disease progression. In this regard, we demonstrate that muscle expression of REG3 proteins is altered after bariatric surgery and is associated with improved muscle insulin sensitivity. The direct systemic activity on insulin resistance, including its impact on muscle glucose uptake and fat/lean mass reduction, as well as the activity in modulating the gut microbiome[17], suggest that REG3A may be a therapeutic approach complementary to existing treatments for type 2 diabetes, such as metformin and GLP1 agonists. This combined with the good clinical safety profile of REG3A (ALF-5755) observed in a phase I study involving healthy volunteers and a phase II study in patients with acute liver failure[23,51], are major arguments for translational exploration of REG3A in patients with type 2 diabetes.

The question of whether REG3A significantly alters metabolism toward weight increase is critical and could undermine the potential clinical uses of REG3A. This was rigorously examined in this study, particularly because previous work reported a spontaneous increase in body weight in mice expressing REG3A in hepatocytes at ages 6-27 weeks[81]. We cast serious doubt on the validity of the Secq et al. study regarding the obesogenic capacity of REG3A by pointing out an internal inconsistency in the data[82] that the authors' response unfortunately did not resolve[83]. To clarify this issue, we followed the weight curve for 24 months of a large number of male and female REG3A transgenic mice and wild-type controls fed a regular standard diet. 125 REG3A transgenic mice (50 males, 75 females) and 102 WT mice (50 males, 52 females) were studied. We show that weight changes are similar between TG-REG3A and WT males over time, and that modest weight differences are observed between transgenic and WT females. Feeding REG3A transgenic and WT mice fatty diets resulted in comparable levels of induced obesity among them. These findings indicate that overexpression of REG3A does not result in an aberrant increase in body weight and are in agreement with those obtained with Reg3b and Reg3g, the murine homologs of REG3A[84].

Overall, this study suggests that REG3A, by mitigating oxidative stress, which is enhanced by metabolic syndromes triggered by Western diets, preserves the function of selective metabolic pathways involved in glucose homeostasis, namely, in this study, gp130-dependent AMPK activation. The antioxidant action of REG3A on the gp130-AMPK pathway does not preclude such a mechanism from triggering the action of other putative REG3A receptors, even though we did not find endogenous expression of EXTL3 in C2C12 muscle cells. Enhanced muscle glucose uptake through REG3A signaling would help reduce insulin requirements in obese, diet-induced insulin-resistant mice. This attenuated hyperinsulinism would ultimately control the progression of insulin resistance and diabetes. Based on the results obtained in mice and in silico, and from a clinical perspective, this study suggests that the administration of a recombinant human REG3A protein may be a valuable approach to reduce peripheral insulin resistance and its associated comorbidities in overweight and type 2 diabetic patients.

## Methods

**Mouse models**. Animal studies were performed in compliance with the institutional and European Union guidelines for laboratory animal care and approved by the Ethics Committee of CE2A-03-CNRS-Orléans (Accreditation Nº01417.01). The number of *Mus musculus* mice used complied with institutional ethical rules and was consistent with common practice in the fields of metabolism studies. Conventional mice were produced and housed in the CNRS SEAT and Institut André Lwoff animal care facilities (Paris-Saclay University, Villejuif). All the REG3A-transgenic mice had circulating human REG3A levels within the 100–500 ng/mL range. Wild-type and REG3A-transgenic mice had the same C57BL/6 N genetic background. The diets used in this study were as follows: standard chow A03 (2830 kcal/kg; 51.7% carbohydrates, 5.1% fat, 21.4% protein), HF235 (4655 kcal/kg; 37.5% carbohydrates, 27.5% fat, 17% protein) or HF260 (5505 kcal/kg; 60% fat, 27% carbohydrates) (SAFE, Augy, France). After 8 weeks on the diet, the mice were first tested for insulin or glucose tolerance and then briefly anesthetized with isoflurane to implant a filled minipump on the right back. Twelve-week-old leptin-deficient male Ob/Ob mice (B6.Cg-Lepob/J) were obtained from Charles River. For rcREG3A administration, an Alzet miniosmotic pump (model 2004) was implanted subcutaneously in 13-week-old mice delivering 9 or 43 µg/day of human recombinant REG3A for 4 weeks. Control groups were infused with an equivalent volume of buffer. Changes in body composition (i.e. fat and lean mass) were determined by quantitative magnetic resonance (EchoMRI 900; Echo Medical Systems, Houston, Texas, USA) in aging mice, in chow and HFD-fed mice and in ob/ob mice on the last day of rcREG3A or buffer treatment (day 29). Body composition was expressed as a percentage of body weight.

**Recombinant human REG3A protein (rcREG3A)**. The rcREG3A protein (ALF5755) is a biological drug that corresponds to the addition of one amino-terminal methionine to the sequence of the secreted (i.e., lacking the 26–amino acid signal sequence) form of the human endogenous REG3A (HIP/PAP) (NP_620355). It was produced in *Escherichia coli*, purified to ≥99% and released in batches in compliance with the clinical grade manufacturing process by PX'Therapeutics (Grenoble, France). The batches of ALF5755 used were supplied by Alfact Innovation.

**Glucose and insulin tolerance tests**. Insulin tolerance tests were performed in mice fasted for 5 h and injected subcutaneously with a rapid-acting insulin (0.5 units/kg or 0.75 units/kg body weight of C57Bl/6N or ob/ob mice respectively; Novorapid flexpen 100 U/mL, NovoNordisk A/S). A glucose tolerance test was performed in mice that were fasted overnight (18 h) and then received a glucose solution at 2 g/kg body weight via oral gavage (glucose 30% CMD Lavoisier). Tail blood was sampled at the indicated time points for glucose and insulin measurements. Blood was immediately centrifuged, and the plasma was separated and stored at −20 °C until assayed. Glucose levels were measured with a glucometer (Glucofix mio, Menarini Diagnostics). Insulin sensitivity was assessed by calculating the slope of the decrease in glucose concentration as a function of time after insulin injection. Surrogate index of insulin resistance (HOMA-IR) was calculated from fasting blood glucose and plasma insulin concentrations as follows: HOMA = (GxI)/405(with glucose (G) expressed as mg/dL and insulin (I) expressed as µU/ml; ref. [85].

**Indirect calorimetry**. The metabolic screening was done using indirect calorimetry. Animals were analyzed for whole energy expenditure (kcal/h), oxygen consumption and carbon dioxide production ($VO_2$ and $VCO_2$, where V is the volume), respiratory exchange ratio (RER = $VCO_2/VO_2$), food intake (g) and locomotor activity (counts/hour) using calorimetric cages with bedding, food and

water (Labmaster, TSE Systems GmbH, Germany). Gas ratio was determined using an indirect open-circuit calorimeter, which monitored $O_2$ and $CO_2$ concentrations by volume at the inlet ports of a tide cage with an airflow of 0.4 L/min, with regular comparisons to an empty reference cage. Whole energy expenditure was calculated according to the Weir equation, using respiratory gas exchange measurements. The flow was previously calibrated with $O_2$ and $CO_2$ mixture of known concentrations (Air Liquide, S.A. France). Animals were individually housed in a cage with lights on from 7 a.m. to 7 p.m. and an ambient temperature of $22 \pm 1\,°C$. All animals were acclimated to their cages for 48 h before experimental measurements. Data regarding food and water consumption were collected every 40 min, and all ambulatory movements recorded during the entire experiment, with the aid of an automated online measurement system combining highly sensitive feeding and drinking sensors and an infrared beam-based activity monitoring system. Gas and movement detection sensors were operational during both light and dark phases, allowing for continuous recording. Animals were monitored for body weight and composition at the beginning and end of the experiment. Data analysis was carried out with Excel XP using extracted raw values of $VO_2$ consumption, $VCO_2$ production (ml/h), and energy expenditure (kcal/h). Subsequently, each value was normalized by whole lean tissue mass extracted from the EchoMRI analysis.

**Calorimetric bomb**. A calorimetric bomb and a fecalogram were performed in the Functional Coprology Laboratory of the Pitié-Salpétrière hospital. Feces were collected over a period of 5 days and 4 nights; feed intake (FI) and fecal output (FO) were recorded. The net absorption balance was calculated as FI-FO, and the percentage of intestinal absorption as 100*(FI-FO)/FI. Dry matter content (dry weight) in g per 100 g of feces (%) was determined by weighing the feces before and after dehydration overnight in an incubator at $+70\,°C$. Total combustion of feces under oxygen allowed calculation of energy leakage via measurement of the amount of heat released. The determination of fecal lipids was performed by a titration technique after extraction of lipids by organic solvents (van de Kamer et al., 1949), and that of proteins by the determination of atomic nitrogen by HPLC. These assays were used to calculate fecal nitrogen and lipid flows and, by subtraction, the carbohydrate flow.

**Quantification of islet insulin content**. The pancreas was fixed in 4% paraformaldehyde overnight, followed by dehydration in graded ethanol solution and embedding in paraffin wax. For each embedded pancreas, five nonconsecutive 5-μm-thick sections (50-μm apart) were mounted on slides and subjected to hematoxylin/eosin staining and anti-insulin primary antibody labeling (RRID:AB_631835; 1 μg/mL). The labeled sections were scanned with Scan 3D Histech P250 Flash III. Quantification of the area of insulin-producing islets was related to the total area of pancreatic sections using QuPath software (version 0.3).

**Hyperinsulinemic-euglycemic clamp**. 5-week-old C57BL/6 N mice were fed chow ($n = 10$) or high-fat ($n = 14$) diet for 10 weeks, and an Alzet miniosmotic pump (model 2004) was implanted subcutaneously in 11-week-old mice delivering 43 μg/day of human rcREG3A ($n = 12$) or buffer treatment ($n = 12$) for the last 4 weeks. Then, the hyperinsulinemic-euglycemic clamp experiments were carried out for conscious mice after a 5 h fast. Insulin perfusions in the jugular vein were at 3mU/minKg body weight and 15 g/dL glucose at variable rate to maintain euglycemia (chow diet: rcREG3A $139.0 \pm 14.0$ mg/dl [$n = 5$], buffer $130.2 \pm 8.9$ mg/dl [$n = 5$]; high-fat diet: rcREG3A $129.7 \pm 12.3$ mg/dl [$n = 7$], buffer $125.2 \pm 8.9$ mg/dl [$n = 7$]). Glucose infusion rates (GIRs) were calculated at the end of an 80-min clamp. To determine [³H]-glucose and 2-[¹⁴C]-DG concentrations, plasma samples were deproteinized with $ZnSO_4$ and $Ba(OH)_2$, dried, resuspended in water, and counted in scintillation fluid for detection of ³H and ¹⁴C. Tissue 2-[¹⁴C]-DG-6-phosphate (2-DG-6-P) content was determined in homogenized samples that were subjected to an ion-exchange column to separate 2-DG-6-P from 2-[¹⁴C]-DG.

**ELISA assays**. The plasma concentrations of REG3A, leptin, insulin, and C-peptide were measured in duplicates using the PancrePAP (Dynabio), mouse leptin ELISA (Crystal Chem; RRID:AB_2722664), mouse ultrasentitive insulin ELISA (ALPCO; RRID:AB_2792981) and mouse C-peptide ELISA (Crystal Chem Inc; Catalog # 90050) kits, respectively, according to the manufacturer's instructions. Plasma concentrations of IL-6, IFN-γ, and TNF-α were measured in triplicate using the ELLA microfluidic platform (Bio-Techne, Minneapolis, MN, USA) according to the manufacturer's instructions.

**Neutral lipid quantification**. Lipids corresponding to 1 mg of liver tissue or 10 μL of plasma of HFD-fed and ob/ob mice were extracted according to Bligh and Dyer[86] in dichloromethane/methanol/water (2.5 :2.5 :2.1, v/v/v), in the presence of internal standards: 4 μg stigmasterol, 4 μg cholesteryl heptadecanoate, 8 μg glyceryl trinonadecanoate. The dichloromethane phase was evaporated to dryness and dissolved in 30 μl of ethyl acetate. 1 μl of lipid extract was analyzed by gas-liquid chromatography on a FOCUS Thermo Electron system using a Zebron-1 Phenomenex fused silica capillary columns (5 m × 0.32 mm i.d, 0.50 μm film thickness). The oven temperature was programmed from 200 °C to 350 °C at a rate of 5 °C per min and the carrier gas was hydrogen (0.5 bar). The injector and the detector were at 315 °C and 345 °C, respectively.

**Cell culture**. Mouse C2C12 myoblasts were grown in a DMEM medium (Gibco) containing inactivated 10% fetal calf serum (FBS), 2 mM Glutamine and 1% penicillin/streptomycin. To induce differentiation, cells were allowed to reach 80–90% confluence in complete medium and then incubated in DMEM with 2% of horse serum. Cells were usually used for experiments 5 days after differentiation. In RNAi experiments, siRNA transfection was performed at day 4 after induction of differentiation using the Viromer Blue transfection kit (Viromer) and a murine GP130- targeting siRNA (ThermoFisher Scientific, RNA ID: MSS236904). The cells were used 24 h after transfection. In experiments performed under oxidative stress conditions, differentiated C2C12 cells were incubated with 10mU of glucose oxidase in fresh DMEM medium with 2% horse serum for 3 h to generate $H_2O_2$ as a by-product of metabolism of medium glucose, and then the cells were lysed. Primary myocytes were maintained in proliferation medium consisting of DMEM-F12 Glutamax (Gibco) supplemented with 20% of inactivated FBS and 2% UltroserG (Pall Life Sciences). The medium was replaced every 2 days. Primary myocytes were seeded at a density of 30,000 cells/cm² on matrigel-coated plates (BD biosciences) in the proliferation medium for 6 h before removing the medium for differentiation medium (DMEM-F12 Glutamax supplemented with 2% inactivated horse serum). Myocytes were kept for 4 days to achieve differentiation. For glucose oxidase treatment, myotubes were serum-starved and incubated for 3 h with glucose oxidase, then switched to serum-free medium containing 100 nmol/l insulin (30 min).

**Immunoblotting**. Whole-cell lysates were made in ice-cold lysis buffer supplemented with phosphatase and protease inhibitors (50 mM Tris-Cl pH7.4, 150 mM NaCl, 1% NP40, 0.25% sodium deoxycholate, 1 mmoL/L $Na_3VO_4$, 20 mmoL/L NaF, 1 μg/mL aprotinin, 10 μg/mL pepstatin, 10 μg/mL leupeptin, 1 μM phenylmethylsulfonyl fluoride). Tissue lysates were obtained from flash-frozen tissue, which was lysed by bead beating in lysis buffer. Protein quantification was performed using the Bio-Rad protein assay kit and bovine serum albumin. 30 μg of protein was resolved in a 12% polyacrylamide gel in Tris-Glycine SDS buffer (Invitrogen), before being electrotransferred onto nitrocellulose membranes (Whatman, Dominique Dutscher). Membranes were blocked in 5% non-fat milk in 0.1% Tween 20 Tris-buffered saline for 1 h and probed with the primary antibodies. AMPK (RRID:AB_915794; dilution 1:1000), Thr¹⁷²P-AMPK RRID:AB_2169396; dilution 1:1000), AKT (RRID:AB_329827; dilution 1:1000), Ser⁴⁷³P-AKT RRID:AB_2315049; dilution 1:1000) and tubulin (RRID:AB_2288042; dilution 1:1000) antibodies were purchased from Cell Signaling Technologies. The gp130 antibody was obtained from Santa Cruz Biotechnology (RRID:AB_647629; 0.2 μg/mL).

**Protein lysate for analysis of oxidative changes**. Differentiated C2C12 cells were lysed in deoxycholate lysis buffer (DOC) containing 2% sodium deoxycholate, benzonase and protease inhibitors (cOmplete Protease Inhibitor Cocktail, Roche), 20 mM Tris-HCl pH 8.8, 2 mM EDTA, 2 mM iodoacetamide. Homogenates were centrifuged at 10,000 g for 10 min at 4 °C. DOC-insoluble fractions were solubilized in SDS lysis buffer containing 1% sodium dodecyl sulfate SDS, 20 mM Tris-HCl pH 8.8, 2 mM EDTA, protease inhibitors, 2 mM iodoacetamide.

**Protein carbonylation**. The carbonyl groups of a DOC-insoluble protein fraction were derivatized to 2,4-dinitrophenylhydrozone (DNPH) by reaction with 2,4-dinitrophenylhydrazine in 0.1% trifluoroacetic acid solution, incubated 15 min in the dark at room temperature. The coupling reactions were stopped by a 1 M Tris solution containing 30% glycerol. Samples were resolved on 10% SDS-PAGE and transferred to nitrocellulose membranes (Whatman, Dominique Dutscher) by electroblotting. Membranes were blocked in 5% non-fat milk in 0.1% Tween 20 Tris-buffered saline for 1 h and probed with a primary antibody against the dinitrophenyl group (DNP) (Millipore, RRID:AB_10850321; dilution 1:50) and then with HRP-conjugated goat anti-rabbit antibody (RRID:AB_390191; dilution 1:2000) to reveal carbonylated proteins using a chemiluminescent reagent (Pierce). To determine the amount of total protein, the same fraction of insoluble DOC protein was resolved on a 10% acrylamide gel for Coomassie blue staining. The amounts of carbonylated and total proteins were determined by densitometry using Image J software; the results were presented as the ratio of carbonylated to total proteins.

**Oxidation of the GP130 and insulin receptor**. DOC-insoluble protein fractions were precipitated with trichloroacetic acid (TCA), protein pellets were washed with ethanol/ethyl acetate solution (50/50) and resuspended 100 mM sodium acetate pH 6.0, 20 mM NaCl, 1% SDS and the carbonyl groups were derivatized by reaction with biotin hydrazide for 2 h before reduction with sodium cyanoborohydride for 30 min at 4 °C. After derivatization, the proteins were precipitated with TCA, the protein pellets were washed with ethanol/ethyl acetate solution (50/50) and resuspended in SDS lysis buffer containing 1% SDS, 20 mM Tris-HCl pH 8.8, 2 mM EDTA, protease inhibitors, and 2 mM iodoacetamide. An aliquot of each sample was frozen for later used as input for Western blotting while the remainder was diluted in equilibration buffer containing 100 mM potassium phosphate, 150 mM NaCl, 400 mM ammonium sulfate pH 7.2. Biotin-derived proteins were captured on a 50 μL streptavidin mutein matrix (Roche), non-specific binding was removed by washing in 100 mM potassium phosphate, 150 mM NaCl pH 7.2, and

biotinylated proteins were eluted in 1X Laemmli loading sample buffer and processed for anti-gp130 and insulin receptor immunoblotting. Input aliquots and eluted protein fractions were resolved on 12% SDS-PAGE and transferred onto nitrocellulose membranes by electroblotting. Membranes were blocked as described previously and probed with streptavidin-biotinylated HRP complex (GE healthcare), rabbit polyclonal gp130 (RRID:AB_647629; 0.2 μg/mL) and insulin receptor β (RRID:AB_631835; 1 μg/mL) antibodies (Santa Cruz Biotechnology), followed by HRP-conjugated goat anti-rabbit antibody (RRID:AB_390191; dilution 1:2000) to reveal proteins using chemiluminescent reagent (Pierce). The amount of proteins was determined by densitometry using Image J software; results were presented as the ratio of carbonylated proteins (biotinylated captured protein) on total proteins (input).

**In silico studies**. Bulk tissue gene expression for REG3A was assessed from the GTEx portal (gtexportal.org; version V8). The Gene Expression Omnibus (GEO) data repository was used to study the expression profile of REG3A in liver, pancreas, white adipose tissue and skeletal muscle of patients with type 2 diabetes. We analyzed 21 publicly available datasets for normalized differential gene expression using limma and edgeR R packages. Enrichment scores were calculated using the ssGSEA algorithm based on the KEGG and Reactome collections of the Molecular Signatures database (MSigDB). Pearson correlation coefficients were calculated between REG3A expression and enrichment scores to determine the relevant pathways associated with REG3A. Local false discovery rates were measured using the Benjamini-Hochberg method. All calculations were performed with the R programming language (v.4.1.3). $P$ values < 0.05 were deemed statistically significant.

**Statistics and reproducibility**. Differences between two sample groups were tested using Student's one-tailed $t$ test or Wilcoxon test. Changes over time (cumulative food intake, energy expenditure, respiratory exchange ratio, fat oxidation, weight gain, OGTT, ITT) were assessed by two-way repeated measures ANOVA followed by Tukey's post-comparison test. Differences between more than two sample groups were tested by ANOVA followed by Tukey's post comparison test when conditions for normality were met. Otherwise, we evaluated by the Kruskal–Wallis test followed by a pairwise Wilcoxon test with Benjamini–Hochberg correction. Results are presented as means ± SEM or box plots. $P$-values < 0.05 were deemed statistically significant. Raw data are available in Supplementary Data and figures.

**Reporting summary**. Further information on research design is available in the Nature Portfolio Reporting Summary linked to this article.

## Data availability

All raw data generated (Supplementary Data 1–7; Supplementary Fig. 12) and their analysis performed during this study are included in this published article (and the related supplementary information files).

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

## Acknowledgements

The authors warmly thank Dr. Bénédicte Chazaud and Dr. Rémi Mounier of the NeuroMyoGène Institute, Lyon, France for fruitful discussions and for providing primary myocytes, Dr. Simone Niclou of the Neuro-Oncology lab, Luxembourg Institute of Health, for providing C2C12 myoblasts. We thank Dr. Bernard Guigui for histological analysis of liver tissues. For experimental assistance, we thank B. Peuteman, G. Duvallet and C. Auriau (animal care), J. Castel and R. Denis (metabolic cages, echoMRI), E. Philippe and J. Denom (glucose clamp), the Metatoul-Lipidomic Facility Core MetaboHUB (Inserm I2MC U 1297, Toulouse, France) and the Functional Coprology Laboratory (Pitié-Salpétrière hospital, Paris, France). This work was supported by grants from the Fondation ARC pour la Recherche sur le Cancer, the National Institute of Health and Medical Research (INSERM) and ALFACT Innovation.

## Author contributions

P.G., A.D.S., M.D., N.M., D.R., C.L., T-S. N., V.S.M., A.D., C.C-G., and C.M. performed experiments. P.G., M.D., N.M., C.L., C.C-G., G.A., P.A., C.B., F.A., C.M., and J.F. designed and conceived the experiments. P.G., A.D.S., M.D., N.M., D.R., C.L., T-S. N., G.A., P.A., C.B., C.C-G., F.A., C.M., and J.F. analyzed experiments. G.A. and P.A. provided the recombinant ALF5755 (REG3A) protein. J.F. wrote the manuscript with contributions from all authors. J.F. supervised the study.

## Competing interests

Paul Amouyal is the Chairman of Alfact Innovation. Gilles Amouyal is CEO of Alfact Innovation. Gilles Amouyal, Paul Amouyal, Christian Bréchot, and Christophe Magnan are co-founders of The Healthy Aging Company (THAC). Patrick Gonzalez, Alexandre Dos Santos, Marion Darnaud, Nicolas Moniaux, Delphine Rapoud, Claire Lacoste, Tung-Son Nguyen, Valentine S Moullé, Alice Deshayes, Céline Cruciani-Guglielmacci, Fabrizio Andréelli, and Jamila Faivre declare no competing interests.
