## [Peer Review File · Communications Biology]

Reviewers' comments:

Reviewer #1 (Remarks to the Author):

The manuscript by Gonzalez P et al demonstrates that overexpressing human REG3A in mouse liver improved insulin sensitivity, and the administration of recombinant REG3A in ob/ob or HFD-induced obese mice improved glucose homeostasis via reducing oxidative protein damage, insulin-stimulated glucose uptake dependent on a gp130-AMPK activation. This manuscript convincingly proves the role of REG3A in glucose homeostasis. However, several concerns need to be addressed.

Major concerns :

- 1.The authors showed that REG3A activated AMPK via gp130 in C2C12cells and concluded that REG3A exerted antioxidant activity most likely via promoting gp130-AMPK signaling. However, it has been reported that EXTL3 is a receptor for REG3A (Lai Y et al, *Immunity*, 2012), would REG3A activate AMPK via EXTL3? The authors need to verify it.
- 2.The authors showed that male REG3A-TG mice have much higher level of REG3A in serum than female REG3A-TG mice. Is the systemic insulin sensitivity in male and female REG3A-TG mice different?
- 3.Why do REG3A-TG mice have lower stimulated blood insulin levels but higher systemic insulin sensitivity?
- 4.In Fig.4g,h, it looks that the administration of 9ug of rcREG3A improves systemic insulin sensitivity better than the administration of 43ug of rcREG3A. Why? Moreover, why was only 43ug of rcREG3A given to HFD-induced prediabetic mice in Fig. 5? The authors need to address it.
- 5.Fig. 6c, the abundance of AKT varies in different lanes. Do different treatments influence the expression of AKT in C2C12 cells? If so, the authors need to add another endogenous control for all western blots.

Minor concerns :

- 1.In Extended Data Fig. 3c and Extended Data Fig. 4a-d, why was the abundance of p-AKT more than that of AKT? Was the exposure time of these two proteins different?
- 2.In Fig.6i and Extended Data Fig. 10c, why was the abundance of p-AMPK more than that of AMPK? Was the exposure time of these two proteins different?
- 3.The authors used Students' t test to analyze the significance of the means of all data. However, Students' t test is usually used for the data with two groups, one categorical variable. Most of the data in this manuscript have more than two groups and two or more categorical variables. Therefore, two-way ANOVA analysis is more appropriate for analyzing the significances of the data.

Reviewer #2 (Remarks to the Author):

This is an interesting novel study that includes a large amount of data on the potential role of REG3A in glucose homeostasis. However, there are a few questions and issues that it would be useful for the authors to address.

- Why did the authors choose to focus almost entirely on male mice? In the text it is stated that females showed significant differences in weight gain at younger ages (4, 6 and 18 months), suggesting a different phenotype to males, potentially more pronounced. It would have been useful to assess female mice as well to assess potential sex differences.
- For the female weight gain data, the text states TG-REG3A vs WT weight gain at 4, 6, and 18 months of age as $21.6 \pm 3.4g$ vs. $22.8g \pm 1.6g$, $P=5.10^{-4}$; $25.2 \pm 6.6g$ vs. $24.7 \pm 2.6g$, $P=0.032$; $28.2 \pm 3.8g$ vs. $30.1 \pm 4.4g$; $P=7.10^{-3}$ respectively. This suggests that at 6 months the TG-REG3A weight was $25.2 \pm 6.6g$ vs. the WT weight of $24.7 \pm 2.6g$, contrary to the text which states that weight gain was lower in TG-REG3A mice?

- Why is the data in Figure 2E given as % baseline when other ITT graphs are provided with actual blood glucose values (e.g. Fig 3F). Presumably this was done because the different animal groups in Fig 2E had very different baselines, but it would be useful to see the original data as with the other graphs.
- Do the authors have any comment on the 24 month old WT mice being insulin resistant compared to 6 month WT mice in the HOMA-IR data (Fig 2D), but not in the ITT data (Fig 2E)?
- Did the authors measure food intake in HFD fed animals? Given that the control diet fed REG3A mice had increased food intake (Fig 1D) was it surprising that there was no difference in weight gain between groups on the HFD (Fig 3A) or do they eat the same amount on a HFD?
- Experimental protocols included in figure 4A and 5A reference CT scans, but these don't appear to be mentioned in the methods section or included in the results?
- In figure 7A the different categories of patients are not clearly defined (e.g. NGT, IGT, etc). What is the difference between normal glucose tolerance (NGT) and insulin sensitive (IS)? Or between IGT and IR?
- The statistics section in the methods is very brief and for most sets of data a t-test has been used. T-tests aren't appropriate tests to make multiple comparisons between groups. For example, in Figure 1A differences between groups have been analysed at each time point and differences between the same group at different time-points have been analysed separately by t-test. The same set of data points should not be included in multiple separate t-test comparisons, the correct test to use here would be an ANOVA. GTT or ITT data over time should be analysed using a repeated measures 2-way ANOVA, etc.

Gonzalez et al. Antimicrobial protein REG3A regulates glucose homeostasis and insulin resistance in obese diabetic mice

Comments from the Editors and Reviewers:

We thank the reviewers and editors for their encouraging and positive feedback on our work and appreciate their constructive comments that helped us improve our manuscript.

Reviewers' comments:

Reviewer #1 (Remarks to the Author):

The manuscript by Gonzalez P et al demonstrates that overexpressing human REG3A in mouse liver improved insulin sensitivity, and the administration of recombinant REG3A in ob/ob or HFD-induced obese mice improved glucose homeostasis via reducing oxidative protein damage, insulin-stimulated glucose uptake dependent on a gp130-AMPK activation. This manuscript convincingly proves the role of REG3A in glucose homeostasis. However, several concerns need to be addressed.

Major concerns :

Comment. 1.The authors showed that REG3A activated AMPK via gp130 in C2C12cells and concluded that REG3A exerted antioxidant activity most likely via promoting gp130-AMPK signaling. However, it has been reported that EXTL3 is a receptor for REG3A (Lai Y et al, Immunity, 2012), would REG3A activate AMPK via EXTL3? The authors need to verify it.

Response. We thank the reviewer for pointing out that exostosin-like 3 (EXTL3), which is involved in the heparan sulfate biosynthetic pathway, would be a player in REG3A signaling. This is an interesting and truly complex issue that requires a full-fledged study for the following reasons:

(i) REG3A is a secreted C-type lectin that regulates a wide range of physiological functions in several different ways. The mechanisms by which the glycan-binding protein REG3A initiates intracellular signaling remain poorly defined, even in light of EXTL3 studies, and involve, depending on the context, tissue and cell type, carbohydrate ligand recognition, carbohydrate-protein interaction, or protein-protein interaction.

(ii) In addition to the EXTL3 glycosyltransferase, other surface receptors and membrane-associated proteins (such as gp130, fibronectin, EGFR, syndecan2, annexin3) have been proposed to interact with REG3A and transmit the REG3A signal into cells.

(iii) Studies on the interaction between EXTL3 and REG3A or Reg1 are insufficiently conclusive and the experimental demonstration is often limited or even contradictory. Also, negative results of interaction between EXT proteins and REG family members were reported by a yeast two-hybrid assay (Mueller et al., Journal of Surgical Research 2008; Chen et al., Front Cell Dev Biol. 2019; Levetan et al., Endocr Pract 2008, Kobahashi et al. JBC 2000 ; Lai et al., Immunity 2012).

(iv) According to the study, silencing of EXTL3 either inhibited REG3A-mediated cell proliferation (Lai et al. 2012) or did not affect the ability of REG3A to modulate it (Liu et al., Cancer Letter 2015), suggesting that EXTL3 may or may not be part of the REG3A cascade.

(v) Finally, the glycosyltransferase activity of EXTL3 is not required for the regulatory action of REG3A on keratinocytes or Reg1 on pancreatic cells which raises questions about the biological significance of the EXTL3/REG3A interaction (Lai et al., Immunity 2012; Yamada S. Cell Mol Biol Lett. 2020; Busse-Wicher et al., Matrix Biology 2014).

We believe that elucidation of the mechanisms of metabolic control by REG3A acting through the putative receptor EXTL3 is beyond the scope of this manuscript. We have addressed the reviewer's comment and added a statement on the issue of putative cellular receptors in the Introduction section of the new version.

Comment. 2.The authors showed that male REG3A-TG mice have much higher level of REG3A in serum than female REG3A-TG mice. Is the systemic insulin sensitivity in male and female REG3A-TG mice different?

Response. Following the suggestion of the reviewer, whom we thank, we examined the insulin sensitivity of female REG3A-TG mice fed a high-fat diet (HF260; 5505 kcal/kg) for 8 weeks. We found that females were more resistant to HFD-induced obesity and metabolic impairment than males, in agreement with the literature (Pettersson et al. PlosOne 2012), and that this better insulin sensitivity of females is comparable whether the genotype is WT or transgenic (new Extended Data Fig. 4). We also found that systemic insulin sensitivity did not correlate with circulating REG3A levels in either the male or female groups, suggesting that changes in circulating levels do not affect the extent of insulin sensitivity control.

Comment. Why do REG3A-TG mice have lower stimulated blood insulin levels but higher systemic insulin sensitivity?

Response. We demonstrate that the human REG3A transgene in non-insulin-dependent mice causes a significant decrease in the HOMA-IR index and an increase in the hypoglycemic effect of exogenous insulin, leading to a greater decrease in blood glucose after intraperitoneal administration of rapid-acting insulin in an insulin tolerance test (ITT). This effect was observed in 6- and 24-month-old transgenic mice. It was not accompanied by an increase in insulin secretion, neither at basal level nor in the OGTT, but on the contrary by a reduction in insulin secretion. This means that REG3A has a specific effect on reducing insulin resistance (or increasing insulin sensitivity) and not an effect on insulin secretion (Figure 2).

A number of more or less related mechanisms could explain the impact of REGA on preventing the pathogenesis of insulin resistance. This study and previous work suggest that several factors, separately or in combination, may contribute to the phenomenon of decreased whole-body insulin resistance in REG3A transgenic mice. Because ectopic lipid accumulation is clearly associated with insulin-resistant states, proper maintenance of insulin homeostasis in aging REG3A mice may be a byproduct of the marked reduction in body fat in these mice. It is conceivable that the observed effect of the REG3A protein on glucose uptake in glycolytic skeletal muscle and not in fat helps counteract excessive triglyceride storage in adipocytes and

explains the decrease in fat mass and increase in lean mass in REG3A transgenic mice over time. We cannot exclude that an effect of REG3A on the gut microbiome may also contribute to the healthy metabolic control of these mice, and only further studies can establish this possibility. We have provided a few sentences in the Discussion section at the request of the reviewer, whom we thank for seeking additional information.

Comment. 4. In Fig. 4g,h, it looks that the administration of 9ug of rcREG3A improves systemic insulin sensitivity better than the administration of 43ug of rcREG3A. Why? Moreover, why was only 43ug of rcREG3A given to HFD-induced prediabetic mice in Fig. 5? The authors need to address it.

Response. We thank the reviewer for pointing out the lack of presentation of the dose comparison for a statistical difference. We found no significant difference in insulin sensitivity (HOMA-IR) of ob/ob mice receiving a dose of 9 μ g or 43 μ g per day at D11 or D25 after rcREG3A infusion (D11: P=0.39; D25: P=0.063, nonparametric anova test) (Figure 4I). Because there was no difference between 9 and 43 μ g per day, it did not seem reasonable to subsequently analyze the effects of both doses, which would have required a larger total number of animals, and we proceeded with the larger dose.

Comment. 5. Fig. 6c, the abundance of AKT varies in different lanes. Do different treatments influence the expression of AKT in C2C12 cells? If so, the authors need to add another endogenous control for all western blots.

Response. Below are the relatively similar b-actin levels between lanes in the experiment shown in old Figure 6c, effectively suggesting that the treatments, most notably the oxidative stress-inducing glucose oxidase treatment (GO, lanes 3, 4, 7, 8), influence total Akt levels. We apologize for submitting an immunoblot that is not representative of the total of 4 anti-total Akt/anti-PhosphoAKT immunoblots performed for densitometry. We replaced this blot (which introduces a level of complexity considering the modulation of AKT expression and activity by oxidative stress reported by many authors) by a new blot and repeated the densitometric measurement on three experiments.

New FIGURE 6c

We made the b-actin in the new figure 6c but did not find it useful to incorporate it into the figure in the new version because of its poor quality.

Minor

concerns :

Comment. 1. In Extended Data Fig. 3c and Extended Data Fig. 4a-d, why was the abundance of p-AKT more than that of AKT? Was the exposure time of these two proteins different?

Comment. 2. In Fig.6i and Extended Data Fig. 10c, why was the abundance of p-AMPK more than that of AMPK? Was the exposure time of these two proteins different?

Response. We first hybridized the membranes with anti-phospho-AKT or phospho-AMPK antibodies, and then the membranes were stripped and reprobbed with anti-total Akt or anti-total AMPK antibodies. Although our stripping procedure is performed under gentles conditions to minimize antigen loss, it is quite possible that the reduction in total protein signal intensity is due to membrane antigen loss. Differences in the affinity of antibodies for their targets may also contribute to this difference in detected signals between total and phosphorylated proteins.

Comment. 3.The authors used Students' t test to analyze the significance of the means of all data. However, Students' t test is usually used for the data with two groups, one categorical variable. Most of the data in this manuscript have more than two groups and two or more categorical variables. Therefore, two-way ANOVA analysis is more appropriate for analyzing the significances of the data.

Response. We apologize for this insufficiency. All required p-values have been recalculated for the data set corresponding to all figures. The description of the statistical analysis procedure is now provided in the Method section of the new version of the manuscript.

Reviewer #2 (Remarks to the Author):

This is an interesting novel study that includes a large amount of data on the potential role of REG3A in glucose homeostasis. However, there are a few questions and issues that it would be useful for the authors to address.

Comment. Why did the authors choose to focus almost entirely on male mice? In the text it is stated that females showed significant differences in weight gain at younger ages (4, 6 and 18 months), suggesting a different phenotype to males, potentially more pronounced. It would have been useful to assess female mice as well to assess potential sex differences.

Response. There is a well-known sex difference in the control of metabolic homeostasis. Like other teams, we generally use male rodents because they are more sensitive to induced metabolic disorders. Following the reviewer's request, we fed females the HFD260 diet for 2 months and, as shown in Extended Data Fig. 4, female mice fed the HFD diet have better glucose tolerance and insulin sensitivity than male mice, regardless of wild-type or transgenic genotype. Therefore, it is impossible to unmask a possible effect of the REG3A transgene on insulin sensitivity in females, at least under these experimental conditions. We are aware that it is necessary to work on both sexes, especially in preclinical research in the therapeutic field, but here our objective was to evaluate the ability of REG3A to influence glucose homeostasis under pathological conditions, so it seemed appropriate to perform work in a model where metabolic disease is clearly established.

Comment. For the female weight gain data, the text states TG-REG3A vs WT weight gain at 4, 6, and 18 months of age as $21.6 \pm 3.4\text{g}$ vs. $22.8 \pm 1.6\text{g}$, $P=5.10^{-4}$; $25.2 \pm 6.6\text{g}$ vs. $24.7 \pm 2.6\text{g}$, $P=0.032$; $28.2 \pm 3.8\text{g}$ vs. $30.1 \pm 4.4\text{g}$; $P=7.10^{-3}$ respectively. This suggests that at 6 months the TG-REG3A weight was $25.2 \pm 6.6\text{g}$ vs. the WT weight of $24.7 \pm 2.6\text{g}$, contrary to the text which states that weight gain was lower in TG-REG3A mice?

Response. Thanks for the comment pointing out this inconsistency and frankly sorry for reversing the weight values. Weights at 6 months are $24.7 \pm 2.6\text{g}$ for TG-REG3A vs. $25.2 \pm 6.6\text{g}$ for WT, $P=0.032$. Change made in the text.

Comment. Why is the data in Figure 2E given as % baseline when other ITT graphs are provided with actual blood glucose values (e.g. Fig 3F). Presumably this was done because the different animal groups in Fig 2E had very different baselines, but it would be useful to see the original data as with the other graphs.

Response. Done. The AUC was calculated from the raw data. Thank you for this remark.

Comment. Do the authors have any comment on the 24-month old WT mice being insulin resistant compared to 6 month WT mice in the HOMA-IR data (Fig 2D), but not in the ITT data (Fig 2E)?

Response. The HOMA-IR score is calculated from fasting blood glucose and insulin levels. Figures 2c and d show a trend toward basal hyperinsulinemia in 24-month-old WT mice compared with 6-month-old mice, which is reflected in the HOMA-IR calculation (not significant $p = 0.17$; post hoc test). This pattern of hyperinsulinemia in aged individuals combined with preserved insulin sensitivity (Figure 2e) is consistent with previous studies in elderly human subjects (Fink et al., Diabetes 1985) and in 18-month-old C57bl/6 mice (Marmentini et al. Frontiers in Endocrinology 2021). These studies demonstrate that insulin clearance decreases with age, which may contribute to age-related hyperinsulinemia, without affecting insulin sensitivity in elderly subjects and would be partly due to a decrease in hepatic expression of CEACAM1 and IDE required for insulin clearance.

Comment. Did the authors measure food intake in HFD fed animals? Given that the control diet fed REG3A mice had increased food intake (Fig 1D) was it surprising that there was no

difference in weight gain between groups on the HFD (Fig 3A) or do they eat the same amount on a HFD?

Response. Thank you for this comment. We measured the food intake of animals fed the high-fat diet and observed no difference in the amount ingested by WT and REG3A transgenic mice throughout the diet (new Figure 3a). A small trend is observed at the end of the experiment (repeated measures Anova, $p=0.68$).

Comment. Experimental protocols included in figure 4A and 5A reference CT scans, but these don't appear to be mentioned in the methods section or included in the results?

Response. Body composition analysis was determined by quantitative magnetic resonance (EchoMRI 900; Echo Medical Systems) and not by CT-scan. We have replaced the label CT-scan in Figure 4A and 5A with the label Echo-MRI.

Comment. In figure 7A the different categories of patients are not clearly defined (e.g. NGT, IGT, etc). What is the difference between normal glucose tolerance (NGT) and insulin sensitive (IS)? Or between IGT and IR?

Response. Cohort labeling was taken from the original studies that generated the transcriptomic data. For the GSEs in Figure 7a for liver, pancreas, and WAT transcriptomes, NGT (normal glucose tolerance) means Hb1Ac less than 6%, IGT (impaired glucose tolerance) means HbA1c between 6 and 6.5%, and T2D (type 2 diabetes) means HbA1c greater than 6.5%. For Figure 7b, IS (insulin-sensitive) and IR (insulin-resistant) were defined by Wu et al. by measuring the glucose disposal rate normalized by lean body mass in skeletal muscle during the hyperinsulinemic euglycemic clamp (Wu et al. Endocrinology 2007). Unfortunately, we cannot specify the differences between NGT/IS or IGT/IR, which were determined by separate indicators (HbA1c versus clamp measurements) by the authors. It is reasonable to assume that NGT and IS define the same population of healthy subjects.

Comment. The statistics section in the methods is very brief and for most sets of data a t-test has been used. T-tests aren't appropriate tests to make multiple comparisons between groups. For example, in Figure 1A differences between groups have been analysed at each time point and differences between the same group at different time-points have been analysed separately by t-test. The same set of data points should not be included in multiple separate t-test comparisons, the correct test to use here would be an ANOVA. GTT or ITT data over time should be analysed using a repeated measures 2-way ANOVA, etc.

Response. This comment was also made by Reviewer#1. We apologize for this shortcoming. All required p-values were recalculated for the data set corresponding to all figures using the appropriate tests. The description of the statistical analysis procedure is now provided in the Method section of the new version of the manuscript.

REVIEWERS' COMMENTS:

Reviewer #1 (Remarks to the Author):

The authors addressed most of my concerns. A key concern remains to be addressed. That is, since multiple membrane-associated protein, including gp130, EXTL3, fibronectin, EGFR, syndecan2 and annexin3, are proposed to interact with REG3A and transmit the REG3A signal into cells, only silencing gp130 is really hard to make a conclusion that REG3A, by protecting gp130 from oxidative damage, releases the gp130-dependent AMPK activation pathway. To support this conclusion, silencing one or more other REG3A-interacting membrane proteins to test REG3A-induced AMPK activation is necessary.

Reviewer #2 (Remarks to the Author):

The authors have addressed all of the issues raised by the reviewers satisfactorily. The minor errors have been corrected, points of discussion in the text have been expanded and clarified and there is some additional data to support the work. The rebuttal to the reviewers points is also thorough and well-argued. As such I have no further concerns.

Gonzalez et al. Antimicrobial protein REG3A regulates glucose homeostasis and insulin resistance in obese diabetic mice

Comments from the Editors and Reviewers:

We thank the reviewers and editors for their interest in our study and for their appreciation of the large amount of data on the potential role of REG3A in glucose homeostasis.

Reviewers' comments:

Reviewer #1 (Remarks to the Author):

Comment. The authors addressed most of my concerns. A key concern remains to be addressed. That is, since multiple membrane-associated protein, including gp130, EXTL3, fibronectin, EGFR, syndecan2 and annexin3, are proposed to interact with REG3A and transmit the REG3A signal into cells, only silencing gp130 is really hard to make a conclusion that REG3A, by protecting gp130 from oxidative damage, releases the gp130-dependent AMPK activation pathway. To support this conclusion, silencing one or more other REG3A-interacting membrane proteins to test REG3A-induced AMPK activation is necessary.

Response. We appreciate reviewer #1's suggestion to study other potential targets of REG3A, but we respectfully disagree. The study of membrane proteins that interact with REG3A is far from satisfactory; it is fragmentary and lacks robust experimental validation, and there is at least some discord among experts as to which interacting proteins are of most biological interest or even as to the importance of the protein-protein interaction in REG3A-mediated signaling. Given the imperfect knowledge in this area of REG3A research, we are concerned about furthering the study by indiscriminately examining a catalog of candidate molecules (not even mentioning the time required to build the tools and perform/reproduce the experiments).

We recognize that of the small number of potential REG3A receptor candidates described in the literature, only glycoprotein-130 (gp130) and EXTL3 are commonly recognized as having metabolic actions, designating them as components that can transduce the REG3A signal to regulate glucose homeostasis.

We performed immunoblotting experiments showing that the glycosyltransferase EXTL3 (involved in the glucose metabolism pathway, and thus a molecule to consider in our framework) is not expressed in C2C12 muscle cells (see below), suggesting that the muscle effect of REG3A on AMPK does not require EXTL3. As such, we thank reviewer #1 for drawing our attention to EXTL3.

In any case, we appreciate the comments from reviewer #1 - taking them into account, we propose to soften the statement by stating in the Discussion section that the antioxidant action of REG3A on the gp130-AMPK pathway does not preclude such a mechanism from

triggering the action of other putative REG3A receptors, even though we did not find endogenous expression of EXTL3 in C2C12 muscle cells.

Immunoblot anti-EXTL3: HuH7: Positive control cells transfected with pCMV-EXTL3.

Reviewer #2 (Remarks to the Author):

Comment. The authors have addressed all of the issues raised by the reviewers satisfactorily. The minor errors have been corrected, points of discussion in the text have been expanded and clarified and there is some additional data to support the work. The rebuttal to the reviewers points is also thorough and well-argued. As such I have no further concerns.

Response. We thank reviewer #2 for his/her kind remarks.